# Biochemical basis for the regulation of biosynthesis of antiparasitics by bacterial hormones

Iti Kapoor[1†‡], Philip Olivares[1,2†§], Satish K Nair[1,2,3*]

[1]Department of Biochemistry, University of Illinois at Urbana Champaign, Urbana, United States; [2]Institute for Genomic Biology, University of Illinois at Urbana Champaign, Urbana, United States; [3]Center for Biophysics and Computational Biology, University of Illinois at Urbana Champaign, Urbana, United States

**Abstract** Diffusible small molecule microbial hormones drastically alter the expression profiles of antibiotics and other drugs in actinobacteria. For example, avenolide (a butenolide) regulates the production of avermectin, derivatives of which are used in the treatment of river blindness and other parasitic diseases. Butenolides and γ-butyrolactones control the production of pharmaceutically important secondary metabolites by binding to TetR family transcriptional repressors. Here, we describe a concise, 22-step synthetic strategy for the production of avenolide. We present crystal structures of the butenolide receptor AvaR1 in isolation and in complex with avenolide, as well as those of AvaR1 bound to an oligonucleotide derived from its operator. Biochemical studies guided by the co-crystal structures enable the identification of 90 new actinobacteria that may be regulated by butenolides, two of which are experimentally verified. These studies provide a foundation for understanding the regulation of microbial secondary metabolite production, which may be exploited for the discovery and production of novel medicines.

*For correspondence:
s-nair@life.uiuc.edu

[†]These authors contributed equally to this work

Present address: [‡]Stanford University, Palo Alto, United States; [§]FDA, Rockville, United States

**Competing interests:** The authors declare that no competing interests exist.

## Introduction

The rise of drug-resistant pathogens continues to compromise human health and is exacerbated by the decline in the rate of discovery of new anti-infectives (*Aminov, 2010*; *Tenover, 2006*). A major limitation is the lack of tools that enable access to the vast library of bacterial natural product antibiotics. Statistical surveys depict the number of antibiotics that are genetically encoded within the *Streptomyces* genus to be in excess of ~300,000 new molecules, but a large repertoire of these compounds cannot be produced when the strain is grown under standard laboratory conditions (*Govern, 2013*; *Watve et al., 2001*). The responsible biosynthetic genes are 'silent' under laboratory conditions and are regulated via unknown mechanisms (*Arakawa, 2018*; *Tyurin et al., 2018*).

The diffusible small molecule γ-butyrolactone (GBL) A-factor plays an essential role in the biosynthesis of the antibiotic streptomycin in *Streptomyces gresius*. The intracellular target of A-factor has been identified as a member of the TetR family, and these receptors have been shown to regulate antibiotic biosynthesis in several actinobacterial species (*Figure 1A*; *Tyurin et al., 2018*; *Takano, 2006*; *Choi et al., 2003*; *Onaka et al., 1995*). The use of exogenous GBLs has been shown to induce secondary metabolite production from otherwise silent clusters (*Thao et al., 2017*; *Sidda and Corre, 2012*). However, the alkali labile nature of γ-butyrolactones and the pleiotropic nature of GBL-mediated regulation limit the general use of these hormones.

Related classes of bacterial hormones include the 2-alkyl-3-methyl-4-hydroxybutenolides, the 2-alkyl-4-hydroxymethylfuran-3-carboxylic acid, and the alkylbutenolides (*Figure 1B*). Butenolides, such as avenolide (*Figure 1B*), trigger the production of secondary metabolites with a minimum

**eLife digest** Bacteria that dwell in soil known as actinobacteria are the source of many drugs that are used to treat cancer and infectious diseases in humans. In their natural environments actinobacteria produce these drugs, or at least similar compounds, to compete with neighboring microbes for food or to kill their enemies. However, when researchers culture actinobacteria in the laboratory, the bacteria often produce little or none of these compounds.

Some actinobacteria produce a compound called avermectin. This compound is closely related to a drug used to treat an infectious disease known as river blindness, which is a common cause of sight loss in people living in West Africa. A bacterial hormone known as avenolide regulates when the bacteria produce avermectin by binding to a receptor known as AvaR1. But the precise details of how this process works remained unclear.

To investigate how avenolide binds to AvaR1, Kapoor, Olivares and Nair developed a new strategy to produce large quantities of avenolide in the laboratory from commercially available molecules. A three-dimensional structure of AvaR1 was then generated showing the receptor on its own, bound to avenolide or bound to a short DNA molecule. In the absence of avenolide, AvaR1 sits on DNA. However, binding to avenolide causes AvaR1 to move off the DNA. This revealed how the binding of avenolide changes the receptor protein so that it can be released from DNA to allow the production and release of other small molecule compounds. Further experiments used these structures as guides to identify 90 new species of actinobacteria that may respond to avenolide and other similar bacterial hormones.

Understanding how bacterial hormones stimulate actinobacteria to produce avermectin and other compounds will aid efforts to extract new compounds from soil bacteria that have the potential to treat cancer or infectious diseases.

effective concentration in the low nanomolar range in *Streptomyces avermitilis* (*Corre et al., 2010*; *Arakawa et al., 2012*). Notably, butenolides show greater pH stability and generally regulate fewer processes than γ-butyrolactones. The recent discovery of avenolide activity in about 24% of observed actinomycetes (n = 51), suggests that other active actinobacteria can also produce avenolide-like compounds to regulate secondary metabolism (*Thao et al., 2017*). For example, avenolide regulates the production of the anthelminitic compound avermectin (*Kitani et al., 2011*; *Miller et al., 1979*). Ivermectin, a chemical derivative of avermectin, is on the World Health Organization's list of essential medicines, and has lowered the incidence of otherwise untreatable parasitic infections, including river blindness, strongyloidiasis, and lymphatic filariasis (elephantiasis) (*Chen et al., 2016*; *Callaway and Cyranoski, 2015*).

Enzymatic and synthetic routes towards the production of γ-butyrolactones have been described, but access to butenolides has been restrictive (*Kato et al., 2007*)., (*Seitz and Reiser, 2005*) Here, we describe a concise 22-step convergent route towards the total synthesis of avenolide, enabling biochemical and biophysical characterization of its interaction with the AvaR1 receptor. We also present structures of AvaR1, in isolation (2.4 Å resolution), in complex with avenolide (2.0 Å resolution), and bound to a synthetic DNA oligonucleotide (3.09 Å resolution) derived from its natural binding site. Using the primary sequence of AvaR1 and the synteny of genes that are likely to be involved in butenolide biosynthesis, we identify 89 additional putative butenolide receptors. Mapping of residues at the ligand-binding site that form conserved sequences highlights their importance in ligand activation. The identification of these putative avenolide-responsive strains may enable the production of novel metabolites in the presence of the hormone. As proof of principle, we show that the supplementation of synthetic avenolide into growing cultures of two strains that contain homologous receptors results in visible changes to the production media.

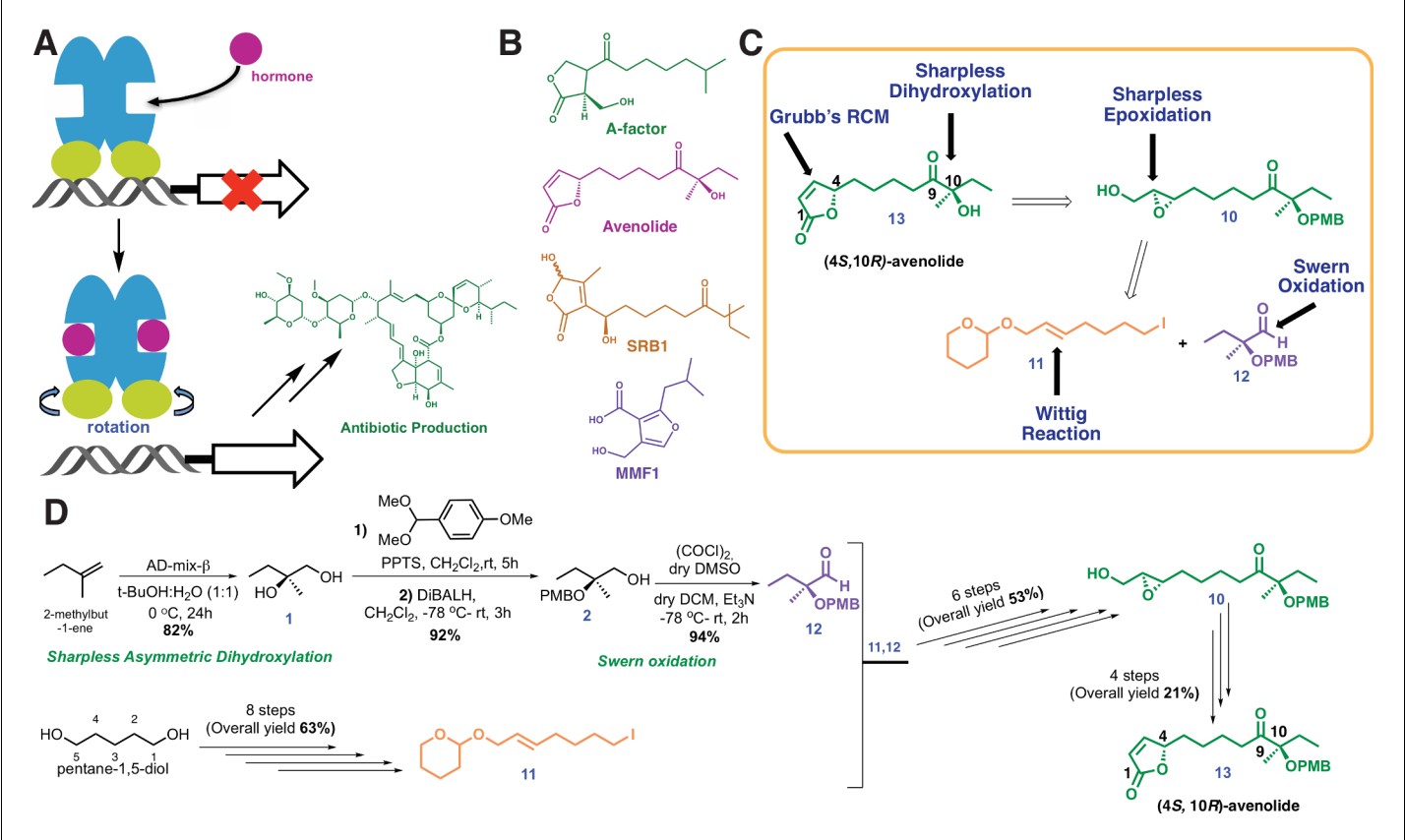

**Figure 1.** Chemical structures and retrosynthetic scheme for avenolide. (A) Representation of the mechanism for hormone-induced transcriptional activation in bacteria. (B) Structures of representative compounds from the four known classes of bacterial hormones. A-factor is a γ-butyrolactone, avenolide is an alkylbutenolide, SRB1 is a 2-alkyl-3-methyl-4-hydroxybutenolide, and MMF1 is a 2-alkyl-4-hydroxymethylfuran-3-carboxylic acid. (C) Retrosynthetic scheme for avenolide synthesis involving five key reactions. (D) Overall summarized and synthetic scheme for total synthesis of (4S,10R)-avenolide with total number of steps and reaction yields.

## Results and discussion

### Convergent synthesis of (4S, 10R)-avenolide and characterization of binding to AvaR1

We hypothesized that a detailed understanding of the mechanism of regulation of secondary metabolite production by avenolide would benefit from an understanding of the regulatory mechanism. Moreover, as AvaR1 shares ~40% sequence identity with the γ-butyrolactone receptor ArpA, biochemical studies of AvaR1 may inform on other classes of bacterial hormone receptors while also providing the rationale for ligand specificity amongst these different classes.

The inherent temperature, acidic and alkali stability of butenolides over γ-butyrolactones prompted efforts towards the large-scale production of avenolide for biochemical studies. However, little is known about the biosynthetic pathways that elaborate butenolides, as opposed to canonical GBLs that can be produced enzymatically from commercially available precursors. To this end, we undertook a novel retro-synthetic strategy produce avenolide from the convergent synthesis of three key fragments: the iodide (**11**), the aldehyde (12) and epoxy (10) (*Figure 1C*).We followed the protocol reported by *Uchida et al., 2011*, which begins with commercially available 1-methyl-2-butene. A Sharpless asymmetric dihydroxylation (*Kolb et al., 1994*) produced the diol intermediate (1) in 82% yield in a single step (*Figure 1D*). The desired key intermediates aldehyde (12), iodo alkene (11) and epoxy (10) were also made following the same reported protocol except that TBDPS rather than TBS was used for improved reaction monitoring using TLC. The epoxy (10) was then stereospecifically converted into allyl alcohol (17) in a single step, by using titanocene dichloride and Zn powder

(*Wang et al., 2013*; *Katsuki and Sharpless, 1980*). The final product, stereospecific (4*S*, 10*R*)-aveno-lide (**13**) was then produced by ring-closing metathesis of molecule **19**, treating the dialkene with Grubb's second-generation catalyst (*Sheddan and Mulzer, 2006*).

The synthetic strategy significantly reduced the number of steps from those reported in prior studies. Specifically, the diol intermediate (**1**) was synthesized from the commercially available 1-methyl-2-butene in single step using Sharpless asymmetric dihydroxylation, avoiding multiple protec-tion/deprotection steps. Likewise, the aldehyde intermediate (**12**) was synthesized effectively in three steps (as compared to ten steps in prior reports), avoiding the use of toxic CuCN. Last, conver-sion of the epoxy intermediate (**10**) to the final product occurred through the intermediacy of an allyl alcohol (**17**), reducing the strategy used in prior reports by four more steps. The final yield of aveno-lide was 14 mg total from 15 g of starting material. The identities of all intermediates, as well as that of the final product, were determined using $^1$H-NMR and $^{13}$C NMR that matched with the reported data. Detailed experimental methods and NMR spectra can be found in Appendix 1.

## Crystal structure of AvaR1 and the binary complex with avenolide

The structure of AvaR1 was determined to 2.4 Å resolution (*Figure 2A*) using crystallographic phases determined from anomalous diffraction data collected from SeMet-labeled protein crystals. The overall structure is reminiscent of that of other TetR-family transcriptional repressors, and consists of an obligate homodimer (*Bhukya and Anand, 2017*). Each monomer is entirely helical and consists of a DNA-binding domain (DBD; composed of α helices 1–4), and a ligand-binding domain (LBD; consisting of α helices 5–13). The dimer interface is formed via interactions between the two LBDs and is formed mainly through hydrophobic packing interactions.

Co-crystallization efforts for AvaR1 bound to avenolide yielded crystals that diffracted to 2.0 Å resolution, and crystallographic phases were determined using molecular replacement (*Figure 2B*; *Thorn and Sheldrick, 2013*). Clear density for the entire hormone can be visualized bound to the LBD of both monomers in the homodimer (*Figure 2C*). Notably, the structure of AvaR1 undergoes conformational shifting upon ligand binding, and a structure-based superposition against the ligand-free structure illustrates that binding of the hormone results in an ~10° shift in the DBD of each monomer (*Figure 2D*). This shift results in an increase in the distance between the two DBD in the dimer upon binding of the ligand, which would preclude DNA binding by the ligand-bound homodimer. Additional local changes between the two structures included the movement of Gln165, which swings into the binding pocket to make hydrogen-bonding interactions with the lac-tone ring, as well with Gln64. Last, Thr108, which is harbored on helix α6, also moves to accommo-date interactions with the hormone. The indole side chain of Trp127 likewise shifts to increase the volume of the binding cavity. Other interactions include hydrogen bonds between Thr131 and the C10 hydroxy, and an interaction between Thr162 and the lactone ring of avenolide. The superimpo-sition suggests that the new contacts formed by these residues are likely to couple hormone binding to the conformational shift between the two monomers of AvaR1 (*Figure 2D*).

We used isothermal titration calorimetry to characterize the binding interaction between the syn-thetic avenolide and AvaR1 (measurements were conducted in triplicate). The resultant binding iso-therm shows the point of inflection at a molar ratio of N-1, which suggests a 1:1 binding of ligand per monomer (*Figure 2D*). The strength of the binding is measured to be $K_d$ = 42.5 nM ± 2.1 nM (three independent trials), which correlates with the reported value of ~4 nM that was estimated from gel shift-based assays (*Kitani et al., 2011*).

## Structure-based comparison across different GBL-like receptors

A structure-based sequence alignment of GBL-like receptors for which ligand specificity has been established reveals a strong conservation of residues that have been shown to be critical for ligand binding, suggesting a common mechanism for ligand recognition across disparate classes of recep-tors (*Figure 3A*). Residues that surround the alkyl chain of avenolide include Trp127, Val158, and Phe161, which are almost universally conserved across all members of the receptor family, whereas Leu88, which forms the opposite wall of the binding cavity, is always a hydrophobic residue but of variable size (*Figure 3B*). The chain length of the methylenomycin furans (MMFs) is shorter than those of the GBLs and butenolides; correspondingly, residues at the base of the ligand-binding cav-ity in the AvaR1, such as Ala85, His130, and Thr131, are replaced by bulkier the Glu107, Leu150,

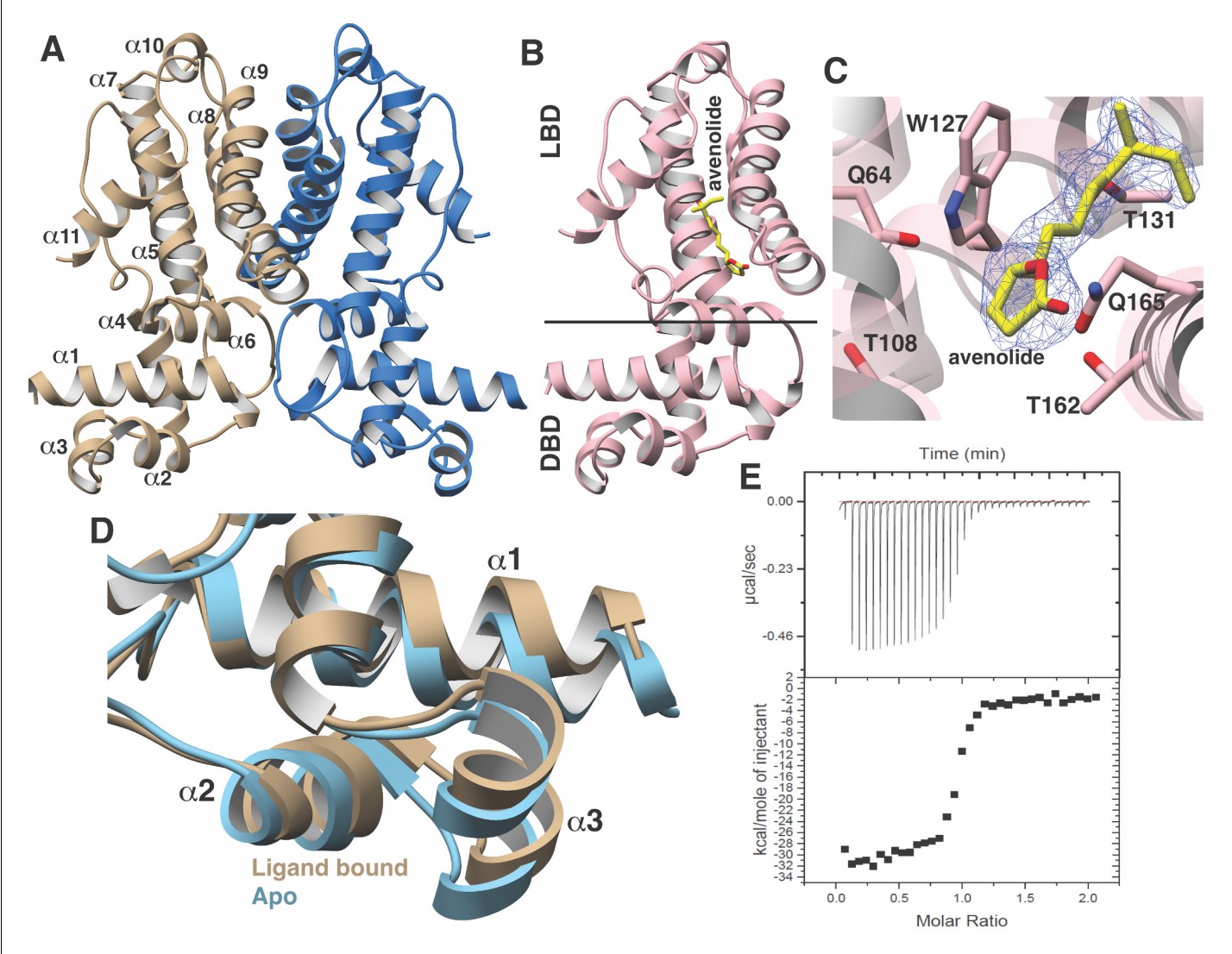

**Figure 2.** Structural characterization of the AvaR1-avenoide binding interaction. (**A**) Structure of the AvaR1 homodimer in the absence of bound ligand. One monomer is shown colored in blue and another in brown. (**B**) Co-crystal structure of one monomer of AvaR1 (in pink) bound to (4*S*,10*R*)-avenolide (in yellow ball-and-stick). The ligand-binding domain (LBD) and the DNA-binding domain (DBD) are indicated. (**C**) Difference Fourier map (countered at 3 σ) calculated with coefficients |F(obs)|–|F(calc)| with the coordinates of the avenolide omitted prior to one round of refinement. The coordinates of the final structure are superimposed. (**D**) Superposition of the structures of the DBD of AvaR1 in the presence (brown) and absence (cyan) of bound ligand. Ligand binding induces a 10° shift in this domain that would preclude DNA binding. (**E**) Representative binding isotherm for the interaction of AvaR1 with (4*S*,10*R*)-avenolide indicative of a 1:1 binding stoichiometry.

and Leu151 in the sequence of the MMF receptor MmrF. Residue Thr161 is within hydrogen-bonding distance to the lactone ring of avenolide, and this residue is conserved in receptors that bind to γ-butyrolactones and butenolides, but absent from receptors for other classes of hormones such as MmrF. Likewise, Gln64 in AvaR1 is positioned on the opposite site of the lactone and is conserved among receptors that bind to structurally related classes of hormones, but is absent from the structure of MmrF. The latter sequence contains a Tyr85 at a near equivalent position, which may be necessary for interactions with the carboxyl group of MMF.

The lactone ring of avenolide is situated above helix α6, which contains residues that are nearly universally conserved among GBL-like receptors, including Ser103, Val104, Arg105, Leu106, Val107, and Asp108. Notably, this helix bridges the LBD and the DBD, suggesting that it plays a role in coupling ligand binding to DNA dissociation (*Figure 3C*). Specifically, movement of Gln64 and Thr108 into the ligand-binding cavity of AvaR1 upon engagement of the hormone results in the

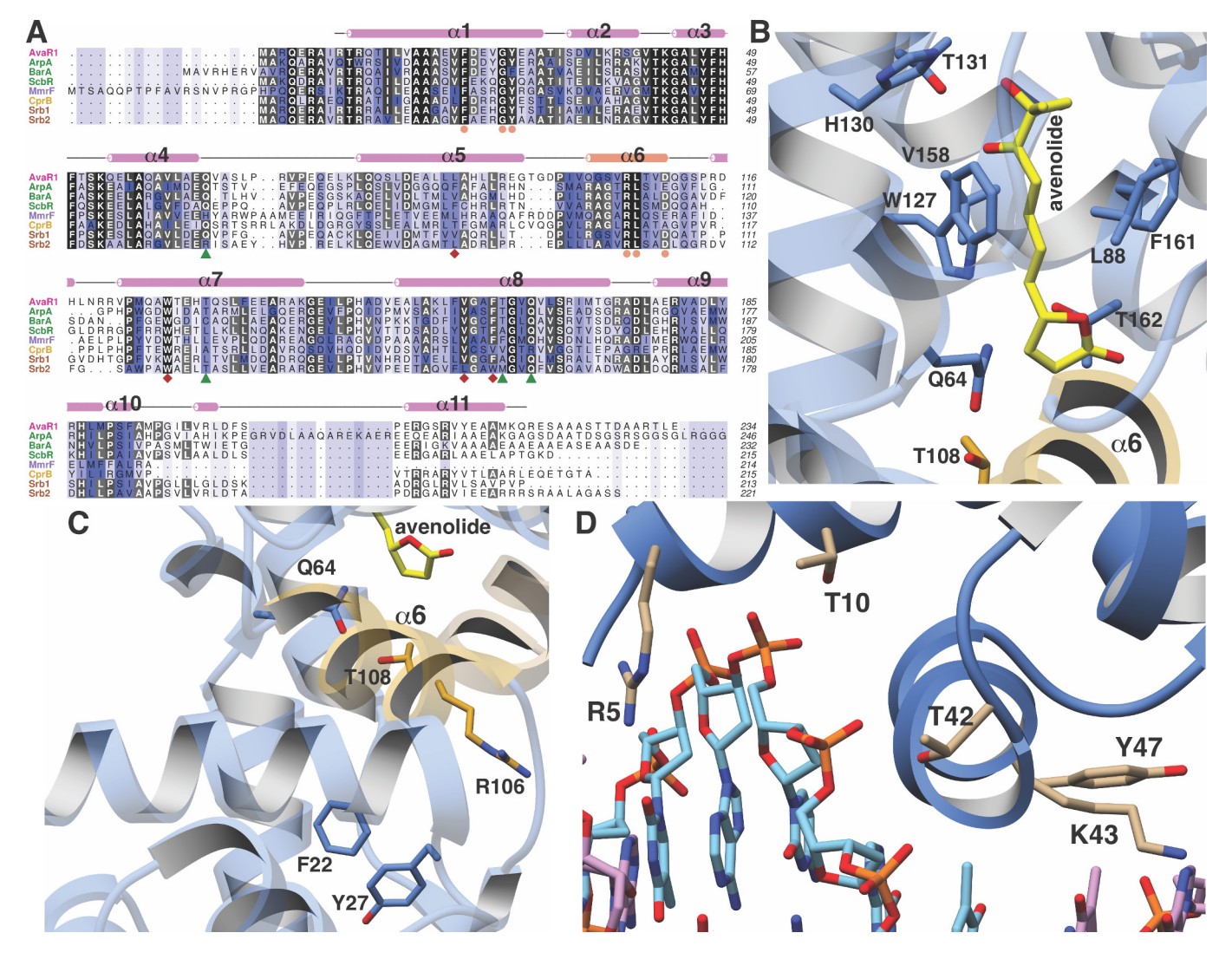

**Figure 3.** Close-up views of AvaR1-ligand and DNA structures. (A) Multiple sequence alignment of various GBL-like receptors for which ligand specificity is known. The color-coding of the receptor names reflects the ligand class as colored in *Figure 1B*. Residues involved in interactions with the lactone are marked by green triangles, those interacting with the alkyl chain are marked by red diamonds, and those proposed to be involved in mediating hormone-dependent conformational movement are marked with orange circles. (B) Close-up view of the hormone-binding cavity showing residues that are in contact with the bound ligand. (C) Spatial orientation of conserved residues that are proposed to induce movement of the DBD in response to binding of the hormone at the ligand-binding domain. (D) Close-up view of the DBD of AvaR1 in complex with the *aco* ARE.

displacement of helix α6 away from the pocket. The orientation of Arg105, located on the opposite side of helix α6, is established through multiple hydrogen-bonding interactions with the backbone carbonyls of conserved residues in helix α1, including the universally conserved Phe22, Gly26, and Tyr27. Hence, the accommodation of hormone binding necessitates movement of the DBD, in order to preserve the suite of hydrogen-bonding interactions with helix α6. As noted, Gln64 and Thr107 are largely conserved among receptors that bind lactone-containing hormones, suggesting a common mechanism for coupling ligand binding to DNA dissociation.

## Crystal structure of AvaR1 and the binary complex with the *aco* ARE

In order to gain further insights into the mechanism of hormone-mediated de-repression, we also determined the structure of AvaR1 bound to a synthetic oligonucleotide derived from the autoregulator responsive element (ARE) sequence. DNase foot-printing analysis had previously established

the identity of the ARE located upstream of the *aco* gene, but this response element is pseudopalindromic (*Kitani et al., 2011*). Crystallization efforts with the symmetric AvaR1 homodimer yielded crystals that did not diffract beyond 8 Å, presumably as a result of the asymmetry of the ARE operator. Efforts using an artificial palindromic sequence derived by inverting and repeating each half of the pseudo palindrome yielded crystals that diffracted to 3.09 Å resolution (*Figure 3D*, *Appendix 1—figure 2*, *Appendix 1—table 1*), and the structure was determined by molecular replacement. As a result of the use of this symmetric DNA, each homodimer in the crystallographic asymmetric unit is bound to a monomer from an adjacent ARE.

The structure shows that each DBD interacts with one half of the palindrome of the DNA duplex. Numerous contacts are formed between helix α1 of the DBD and the duplex, including Arg5, which inserts into the major groove and interacts with Thy7, as well as between Lys43 and Ade15 of the ARE (*Figure 3D*). Additional non-specific interactions include those between Thr10, Thr42, and Tyr47 and the backbone phosphate of the duplex. A comparison of the ligand-binding sites with that in the hormone-bound structure reveals that the binding pocket is further occluded through movements of Trp127 and the loop harboring Gln64, consistent with the roles of these residues in effecting conformational movements of the DBD upon binding of the hormone.

## Genome mining informs on putative butenolide regulatory biosynthetic pathways

Given the improved stability of butenolides over γ-butyrolactones, we speculate that these hormones may prove more amenable in attempts to activate antibiotic production. In order to identify actinobacterial strains that are under butenolide regulatory control, we sought to use a bioinformatics approach based on the identification of the corresponding receptor. However, as shown in *Figure 3A*, the sequence similarity between bona fide γ-butyrolactone receptors, such as ArpA, and butenolide receptors is high (40% sequence identity with AvaR1) precluding such analysis. Orphan receptors called pseudo γ-butyrolactone receptors that are activated by multiple ligands likewise share 40% sequence identity with AvaR1, confounding simple sequence-based analysis. Prior efforts to discriminate between receptor classes have not proven fruitful, and phylogenetic analyses have failed to discriminate between positive and negative regulators.

In an effort to identify other actinobacteria that are under butenolide regulatory control, we utilized synteny of the putative butenolide biosynthetic genes to distinguish between receptor clades. We first used the Enzyme Similarity Tool (EST) from the Enzyme Function Initiative to create a Sequence Similarity Network (SSN) of all members of the receptor class. An E-value cutoff of $10^{-70}$ was used to produce an SNN in which characterized receptors of the various GBL families were segregated with mutually exclusive co-localization (*Figure 4A*). We used the resultant SSN as input for the EFI Genome Neighborhood Network (GNN) tool to identify nodes that are co-localized next to genes with PFams that are associated with putative avenolide biosynthetic genes. The fact that the butenolide receptors often regulate the production of their own ligand further enabled this approach and allowed for inferences about the classes of ligands that are produced and recognized by receptors that have yet to be characterized. Because the EFI tools are only integrated with the Uniprot database, we also manually searched sequences in Genbank for similar operonic architecture to identify putative butenolide receptors on the basis of genomic context.

Our analysis yielded a set of 90 putative actinobacterial genomes (*Appendix 1—table 2*) that harbor a putative butenolide receptor that is likely to be under butenolide regulatory control (*Figure 4A*). We emphasize that, although this is orders of magnitude greater than the currently known number of butenolide receptors, this number represents a significant underestimate but the true number cannot be identified given the limitations of our approach. Mapping of the sequence conservation amongst these 90 receptors onto the co-crystal structure of hormone bound to AvaR1 reveals that residues Gln64, Thr108, and Trp127 , which are proposed to couple ligand binding to domain shift, are conserved among all of these sequences (*Figure 4B*). The conservation score is highest for residues that are involved in interactions with the lactone, whereas those that interact with the alkyl tail are more divergent. These data are consistent with the observations that the lactone ring and C10 hydroxyl are general features of butenolides, whereas the length and branching of the alkyl tail vary significantly. The organization of the biosynthetic operon that harbors genes, which is hypothesized to be involved in butenolide biosynthesis, is largely conserved in these organisms (*Figure 4C* and *Appendix 1—table 2*).

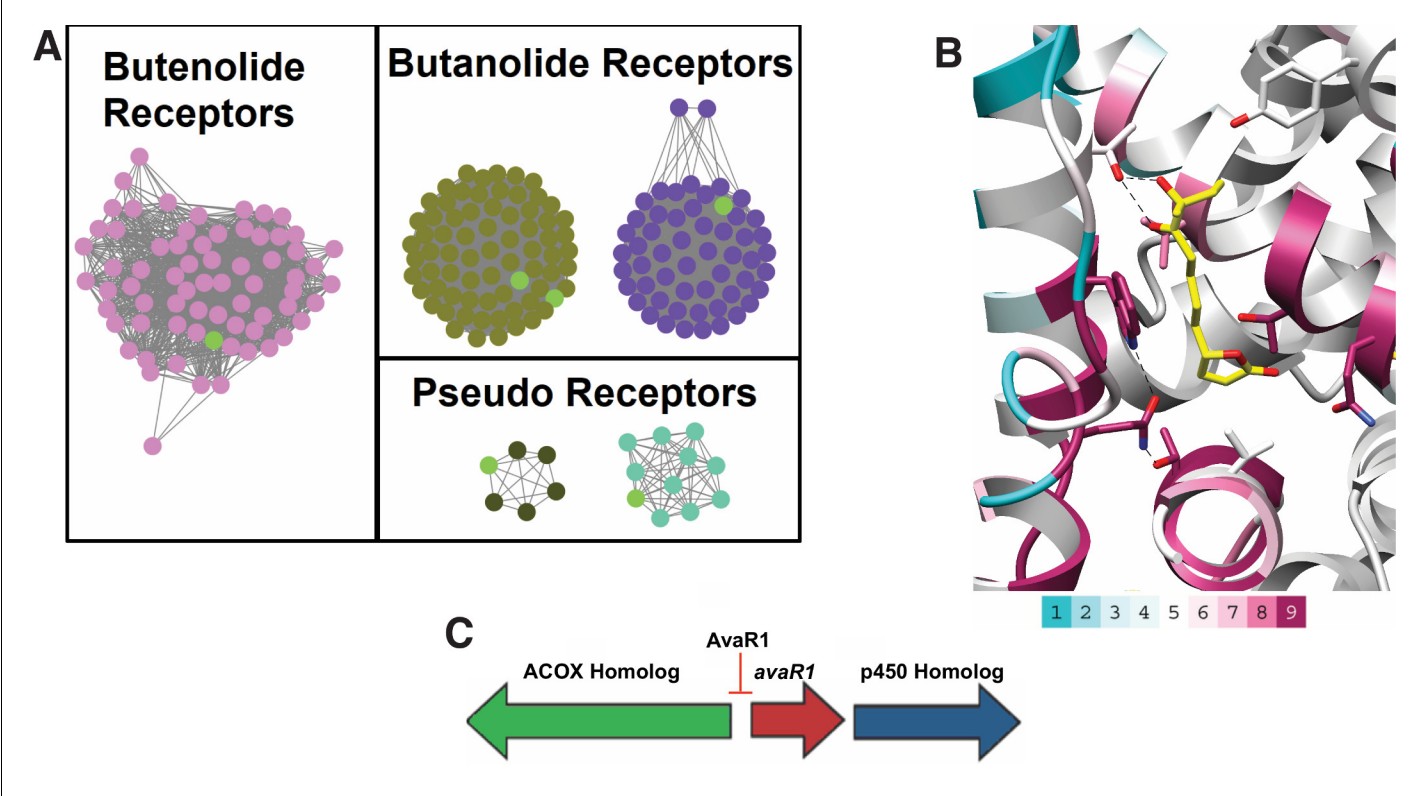

**Figure 4.** Sequence Similarity Networks of likely butenolide gene clusters. (**A**) Sequence Similarity Networks (SSN) showing the relationship between different clades of putative butenolide receptors. Characterized receptors are shown in light green. (**B**) Conservation of sequences amongst the 90 putative butenolide receptors identified by bioinformatics mapped onto the structure of AvaR1. The color range indicates the least conserved (cyan) through to the most conserved (purple). (**C**) Genomic synteny used to cull sequences for the SSN.

## Conclusion

Although the bacterial hormone A-factor was discovered nearly a half century ago, significant gaps remain in our understanding of how these signaling molecules regulate gene expression. The discovery of the regulation of secondary metabolite biosynthesis by γ-butyrolactones inspired efforts to use these molecules as ex vivo effectors to induce otherwise silent biosynthetic gene clusters, but these efforts met with little success. The labile nature of the lactone ring, as well as the often pleiotropic effects that γ-butyrolactones induce, have presumably subverted efforts to utilize these small molecules as chemical inducers. By contrast, the structurally related butenolides show improved stability under strongly acidic and basic conditions. However, the utility of butenolides in biotechnology efforts has been limited by an inability to access these molecules.

Here, we present here an efficient and convergent 22-step total synthetic route for the production of avenolide, which can be extrapolated for the total synthesis of other members of the butenolide class of small signaling molecules. This effort allowed for detailed structure–function studies of the corresponding hormone receptor, including the first crystal structure of any GBL-type receptor bound to its cognate ligand. The structural data informs on the mechanism by which hormone binding induces a conformational change in the AvaR1 receptor, resulting in the formation of a dimeric assembly that occludes efficient DNA binding. We also elaborate a bioinformatics strategy using the genomic neighborhood context of butenolide biosynthetic genes as a marker to identify 90 actinobacterial strains that are likely to be under the regulatory control of butenolides. The addition of avenolide to the growth media for two representative strains results in changes in the color of the culture supernatant. These results support the validity of our bioinformatics approaches and set the framework for further efforts towards the use of butenolides to active antibiotic biosynthesis in otherwise silent gene clusters.

## Materials and methods

Total synthetic schemes, experimental procedures, and validation of relevant synthetic intermediates are provided in the 'Supplemental data'.

### Expression, purification, crystallization and structure determination of AvaR1

Wild-type protein AvaR1 was amplified from *S. avermitilis* genomic DNA by PCR using primers that were based on the published sequence of the polypeptide and inserted into a pET-28-MBP vector for expression in *Escherichia coli* as a maltose binding protein (MBP)-tagged fusion. The resultant plasmid was transformed into *E. coli* containing the Rosetta plasmid for protein expression. AvaR1 was produced by growing the cells in shaking flask of LB media at a temperature of 37°C. When the cells reached an O.D.$_{600}$ of 0.6, the cells were cooled on ice for 15 min. Following the addition of 0.5 mM isopropyl β- d-1-thiogalactopyranoside (IPTG), the cells were placed in an 18°C shaking incubator for 18 hr. The cells were then harvested by centrifugation and re-suspended in a buffer composed of 500 mM NaCl, 20 mM Tris base (pH 8.0), and 10% glycerol. Re-suspended cells were lysed by homogenization and the lysate was centrifuged at 14,000 rpm to remove cell debris. The cleared cell lysate was loaded onto a HisTrap column, which was subsequently washed with 1 M NaCl, 30 mM imidazole, and 20 mM Tris base (pH 8.0). MBP-tagged AvaR1 was eluted using a linear gradient beginning with 1 M NaCl, 20 mM Tris base (pH 8.0), and 30 mM imidazole and ending with 250 mM imidazole. Pure fractions, as judged by SDS-PAGE, were combined and diluted two-fold before treatment with thrombin (final ratio of 1:100 [w/w]) for 18 hr at 4°C to cleave the N-terminal tag. Tag-free AvaR1 was concentrated and loaded onto a size exclusion column (Superdex S75 16/60) pre-equilibrated with 100 mM KCl and 20 mM HEPES free acid (pH 7.5). Pure fractions were collected and concentrated to 25 mg/mL before storage in liquid nitrogen. Production of SeMet-labeled AvaR1 was carried out by repression of methionine synthesis in defined media supplemented with selenomethionine (*Doublié, 2007*).

Preliminary crystals of AvaR1 were obtained using a sparse matrix screen. Diffraction-quality crystals were grown using hanging drop vapor diffusion. 13.5 mg/mL AvaR1 was added at a 1:1 ratio to mother liquor containing 12% polyethylene glycol (PEG) 1000, 0.1 M sodium citrate tribasic dehydrate (pH 4.2), 0.2 M LiSO$_4$ and 4% (v/v) tert-butanol, and incubated against the same solution at 4°C. Crystals were improved through multiple rounds of micro-seeding. The SeMet-AvaR1 crystals were obtained using 12 mg/mL protein added to a 1:1 ratio of 30% PEG MME 2000, and 0.15 M KBr. Crystals were vitrified by direct immersion without the addition of any cryo-protectives.

All diffraction data were collected at Argonne National Laboratory (IL). The autoPROC (*Vonrhein et al., 2011*) software package was utilized for the indexing and scaling of the diffraction data. Initial phases for AvaR1 were obtained using anomalous diffraction data collected on crystals of SeMet-labeled protein. Initial models were built using Phenix and Parrot/Buccaneer. Manual refinements were completed by the iterative use of COOT (*Emsley et al., 2010*) and Phenix.refine. Cross-validation was utilized throughout the model-building process in order to monitor building bias. The stereochemistry of all of the models was routinely monitored using PROCHECK. Crystallographic statistics are provided in *Appendix 1—table 3*.

For co-crystallization of the hormone-bound complex, purified AvaR1 (14 mg/ml) was incubated with 3 mM avenolide for 30 min on ice. Co-crystals were obtained by vapor diffusion methods and initial crystals were obtained in Index D5 (25% PEG3350 and 0.1M sodium acetate trihydrate [pH 4.5]). Well-diffracting crystals were produced through optimization to a final solution of 23% PEG3350 and 0.1 M sodium acetate trihydrate (pH 4.5) at 4°C using hanging drop crystallization. Crystals were submerged briefly in the crystallization medium supplemented with 25% ethylene glycol prior to vitrification in liquid nitrogen. The coordinates of apo AvaR1 were used to determine crystallographic phases.

AvaR1 was co-crystallized with different oligonucleotide sequences that were designed on the basis of the AvaR1 DBS upstream of *aco* gene. Purified and concentrated dimeric protein (14 mg/mL) was incubated with individual oligonucleotide duplexes (Supporting *Appendix 1—table 1*) in 1:1.2 molar ratio, for 30 min on ice. Palindromic DNA sequences were first self-annealed and then double stranded DNA was used from 1 mM stock prepared in 20 mM MgCl$_2$, 50 mM Tris (pH 8.0) buffer. The order in which reactants were added was: buffer, DNA and finally protein. For some

oligonucleotides, white turbid solution was obtained as soon as the protein was added, but the addition of a few microliters of ammonium acetate and incubating at either room temperature or on ice produced clear solution. Crystallization trays were set up at 4˚C and every oligonucleotide was crystallized in different conditions (*Appendix 1—figure 2B*). Ethylene glycol (25% v/v) was used as cryoprotectant prior to the vitrification of crystals for all AvaR1–oligonucleotide co-crystals. The oligonucleotide sequences used are listed in *Appendix 1—table 1* and the sequences that produced diffraction-quality crystals are shown in *Appendix 1—figure 2*.

## Identification of putative butenolide biosynthetic clusters

Using the AvaR1 amino-acid sequence as a handle, tools from the Enzyme Function Initiative (EFI) were used to first create a Sequence Similarity Network (SSN) of 10,000 Uniprot sequences. Once an SSN was created, an iterative process was undertaken to find an E-value that ensured that characterization of receptors of the various GBL families resulted in mutually exclusive co-localization. The resulting E-value was $10^{-70}$. Using this SSN, the data was run through the EFI's Genome Neighborhood Network (GNN) webtool. Network visualization was performed in Cytoscape (*Shannon, 1971*). Using Pfams associated with putative avenolide biosynthetic genes along with the knowledge that this family of receptors often regulates their own ligand production, inferences were made as to the class of ligand produced and recognized by uncharacterized receptors. Because the EFI webtools are only integrated with the uniprot database, we also manually scoured through a number of Genbank sequence results derived from BLAST analysis for proper genomic context relative to butenolide production. These BLAST searches used the sequences of any putative butenolide biosynthetic genes as handles. The proper genomic context necessary for butenolide biosynthesis was defined as a TetR_N Pfam receptor surrounded by a gene in the p450 Pfam (PF00067), and either an Acyl-CoA_dh_1 (PF00441) Pfam gene, an Acyl-CoA-dh_2 (PF08028) Pfam gene, or an ACOX (PF01756) Pfam gene. A list of the 90 homologous strains that were identified is provided in *Appendix 1—table 2*.

## Isothermal titration calorimetry

ITC measurements were performed at 25˚C on a MicroCal VP-ITC calorimeter. A typical experiment consisted of titrating 7 μL of a ligand solution (80 μM) from a 250 μL syringe (stirred at 300 rpm) into a sample cell containing 1.8 mL of AvaR1 solution (8 μM) with a total of 35 injections (2 μL for the first injection and 7 μL for the remaining injections). The initial delay prior to the first injection was 60 s, with reference power 10 μCal/s. The duration of each injection was 16 s and the delay between injections was 400 s. All experiments were performed in triplicate. Data analysis was carried out with Origin 5.0 software. Binding parameters, such as the dissociation constant ($K_d$), enthalpy change ($\Delta H$), and entropy change ($\Delta S$), were determined by fitting the experimental binding isotherms with appropriate models (one-site binding model). The ligand stock solution was prepared at 10 mM. The buffer solutions for ITC experiments contained 300 mM KCl and 20 mM HEPES (pH 7.5).

## Total synthesis of avenolide

The experimental procedures were adopted from *Uchida et al., 2011* with further optimizations and modifications as stated. Detailed experimental procedures for relevant intermediates are specified in the Materials and methods section of Appendix 1.

**Chemical structure 1.** One step synthesis of (*R*)-2-Methylbutane-1,2-diol (**1**) from 2-methyl-1-butene via Sharpless asymmetric dihydroxylation.

For synthesis of (R)−2-Methylbutane-1,2-diol (**1**): To a stirred solution of the 2-methyl-1-butene (3.75 g, 53.47 mmol) in t-BuOH: $H_2O$ (1:1, 400 ml) were added $K_3Fe(CN)_6$ (52.81 g, 160.41 mmol), $K_2CO_3$ (22.17 g, 160.41 mmol), $K_2OsO_4(OH)_4$ (197 mg, 0.54 mmol, 1 mol%) and $(DHQD)_2PHAL$

(416.5 mg, 0.54 mmol, 1 mol%) were added to a stirred solution of 2-methyl-1-butene (3.75 g, 53.47 mmol) in t-BuOH: $H_2O$ (1:1, 400 ml) at 0°C under Ar atmosphere. The reaction mixture was stirred for 24 hr at 0°C using an Ar balloon. The reaction was quenched with a saturated aqueous solution of $Na_2S_2O_3$ and the aqueous phase was extracted with EtOAc (2 × 1 L). The water layer was thoroughly washed with EtOAc, and combined organic extracts were washed with brine (saturated NaCl) and dried over anhydrous $Na_2SO_4$ and concentrated in vacuo. The residue was purified by flash column chromatography with a gradient from 30% EtOAc/hexanes to 70% EtOAc/hexanes to 10% MeOH/DCM, to afford **1** (3.5 g, 82%) as a colorless oil. This molecule was obtained in single step from commercially available 2-methylbut-1-ene using Sharpless asymmetric dihydroxylation (*Kolb et al., 1994*). Spectroscopic characterization parameters agreed with the reported mass and chemical shift values.

[α] (*Bhukya and Anand, 2017*) +4.96 (c 1.0, CHCl₃); ¹HNMR (500 MHz, CDCl₃) δ 3.47 (brs, 2H), 3.43 (d, J = 11.1 Hz, 1H), 3.37 (d, J = 11.1 Hz, 1H), 1.51 (q, J = 7.3 Hz, 2H), 1.10 (s, 3H), 0.89 (t, J = 7.6 Hz, 3H); ¹³C-NMR (125 MHz, CDCl₃) δ 73.6, 69.3, 31.2, 22.5, 8.2 HRMS (ESI+, TFA-Na) calcd for $C_5H_{12}NaO_2$ 127.0735 [M+Na]⁺, found m/z 127.0740.

For synthesis of aldehyde fragment **12**, the p-methoxybenzyl (PMB)-protected intermediates were synthesized according to the protocol reported by *Uchida et al., 2011*. Iodo alkene fragment **11** was also synthesized following the reported protocol by substituting the use of the tert-butyl (dimethyl)silyl (TBS) protecting group with a tert-butyldiphenylsilyl group (TBDPS) to enable easy monitoring of the reaction by thin layer chromatography (TLC) visualization under UV light. Epoxy fragment **10** was synthesized through intermediates **14**, **15** and **16**, as described in Appendix 1.

**Chemical structure 2.** Single step synthesis of allyl alcohol **17** from epoxy **10**.

(R)−8-((2S,3S)−3-(Hydroxymethyl)oxiran-2-yl)−3-((4-methoxybenzyl)oxy)−3-methyloctan-4-one (**17**) was synthesized following the protocol from the reported literature (*Wang et al., 2013*). Anhydrous $ZnCl_2$ (2 mL, 1 M in Et₂O, 2 mmol) and zinc powder (350 mg, 6.72 mmol) were added to a red solution of $Cp_2TiCl_2$ (1.26 g, 5.05 mmol) in anhydrous tetrahydrofuran (THF) (15 mL). The solution was stirred for 1 hr at room temperature until it turned green. Epoxide **10** (590 mg, 1.68 mmol) in anhydrous THF (5 mL) was then added to the resultant mixture. After stirring for 30 min at room temperature, the reaction was quenched with aqueous HCl (1.0 M, 3 mL) and the mixture was extracted three times with Et₂O (4 mL). Collected Et₂O fractions were combined and washed with water, 10% aqueous NaHCO₃, water and brine, dried over $Na_2SO_4$ and filtered and concentrated under reduced pressure. The obtained residue was purified using flash column chromatography on silica gel (25% EtOAc/hexane to 40% EtOAc/hexane) to obtain pure allyl alcohol compound **17**.

¹H-NMR (500MHz, CDCl₃) δ 7.27 (d, J = 8.8 Hz, 2H), 6.89 (d, J = 8.8 Hz, 2H), 5.87–5.81 (m, 1H), 5.21 (ddd, J = 17.2, 1.4, 1H), 5.10 (ddd, J = 10.4, 1.4, 1H), 4.33 (d, J = 10.8 Hz, 1H), 4.29 (d, J = 10.8 Hz, 1H), 4.14–4.06 (m, 1H), 3.81 (s, 3H), 2.66 (dt, J = 7.3, 4.3 Hz, 2H), 1.84–1.70 (m, 2H), 1.59–1.34 (m, 6H), 1.33 (s, 3H), 0.84 (t, J = 7.5 Hz, 3H); ¹³C-NMR (500MHz, CDCl₃) δ 215.2, 159.0, 141.3, 128.9, 128.9, 114.9, 114.0, 113.8, 84.8, 73.2, 65.3, 55.5, 37.1, 36.8, 29.4, 25.3, 23.5, 20.2, 8.1; HRMS (ESI+, TFA-Na) calcd for $C_{20}H_{30}NaO_4$ 357.2042 [M+Na]+, found m/z 373.2032.

This resulted in a simplified protocol for the synthesis of allyl alcohol **17**, which otherwise was reported to be made in two additional reaction steps starting from epoxy **10**. An acrylic group was added to allyl alcohol **17** using DDQ by the reported procedure, which was followed by ring-closing metathesis reaction to yield stereospecific (4S, 10R)-avenolide(13). Detailed synthetic schemes, experimental procedures and yields are reported in the Materials and methods section of Appendix 1. The obtained ¹H and ¹³C NMR data support the reported values, and thus the spectra are provided only for key intermediates.

## Acknowledgements

We thank Keith Brister and colleagues at LS-CAT (Argonne National Labs) for facilitating X-ray data collection. This work is supported in part by NIH grant GM131347 to SKN.

## Additional information

### Funding

| Funder | Grant reference number | Author |
|---|---|---|
| National Institutes of Health | GM131347 | Iti Kapoor |

The funding agency provided support for facilities and services. The agency also provided partial salary support for all participants.

### Author contributions

Iti Kapoor, Validation, Investigation, Methodology, Writing - original draft, Writing - review and editing; Philip Olivares, Data curation, Software, Formal analysis, Validation, Investigation, Visualization; Satish K Nair, Conceptualization, Formal analysis, Supervision, Investigation, Visualization, Writing - original draft, Project administration, Writing - review and editing

### Author ORCIDs

Philip Olivares (ID) https://orcid.org/0000-0003-3182-5884
Satish K Nair (ID) https://orcid.org/0000-0003-1790-1334

### Decision letter and Author response

Decision letter https://doi.org/10.7554/eLife.57824.sa1
Author response https://doi.org/10.7554/eLife.57824.sa2

## Additional files

### Supplementary files

- Supplementary file 1. Key resources table.

- Transparent reporting form

### Data availability

Diffraction data has been deposited in the PDB under accession codes 6WP7, 6WP9 and 6WPA.

The following datasets were generated:

| Author(s) | Year | Dataset title | Dataset URL | Database and Identifier |
|---|---|---|---|---|
| Kapoor I, Olivares P, Nair SK | 2020 | Structure of AvaR1 | https://www.rcsb.org/structure/6WP7 | RCSB Protein Data Bank, 6WP7 |
| Kapoor I, Olivares P, Nair SK | 2020 | Structure of AvaR1 bound to avenolide | https://www.rcsb.org/structure/6WP9 | RCSB Protein Data Bank, 6WP9 |
| Kapoor I, Olivares P, Nair SK | 2020 | Structure of AvaR1 bound to DNA half-site | https://www.rcsb.org/structure/6WPA | RCSB Protein Data Bank, 6WPA |

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

## Appendix 1

### Supplementary methods

All commercially available chemicals were purchased by the Aldrich Company and Fischer Scientific and were used without further purification unless otherwise specified. Anhydrous solvents were either purchased from Fischer scientific and Aldrich or borrowed from solvent dehydrating system (SDS) in other labs at UIUC. All of the oligonucleotides used in the study were purchased from Integrated DNA Technology (IDT). IPTG and antibiotics were purchased from Gold Biotechnology.

### Instrumentation

High- and low-resolution mass spectra were obtained by the Mass Spectrometry Laboratory, School of Chemical Science, University of Illinois. Mass spectra were obtained by field desorption (FD) on a Waters 70-VSE-A and by ESI on a Waters Micromass Q-Tof. Other instruments like rotatory evaporator (Isotemp) and stirring plates (Cole Palmer) were purchased from Fischer Scientific.

Nuclear magnetic resonance (NMR) spectra were recorded on a Varian Unity 500, Varian INOVA 500NB or Varian Unity 400 spectrometer at $21 \pm 3°C$ unless otherwise mentioned. Chemical shifts ($\delta$) are reported in parts per million (ppm). Coupling constants (J) are reported in Hertz. $^1H$ NMR chemical shifts were referenced to the residual solvent peak at 7.26 ppm in chloroform-d ($CDCl_3$) or other deuterated solvents as stated. $^{13}C$ NMR chemical shifts were referenced to the center solvent peak at 77.16 ppm for $CDCl_3$. Analytical thin-layer chromatography (TLC) was performed on 0.2 mm silica 60 coated on glass plates with F254 indicator for monitoring all of the reactions. PMA, $KMnO_4$, anisaldehyde, ninhydrin, and iodine were used for staining TLC plates to detect various functional groups. Flash column chromatography was performed on 40–63 µm silica gel ($SiO_2$). Solvent mixtures used for chromatography are reported as percent volume (%) or as volume ratio (v/v).

### Total synthetic schemes and experimental procedures

The experimental procedures were adopted from the publication by *Uchida et al., 2011* with further optimizations and modifications as described.

**Appendix 1—chemical structure 1.** Chemical structures and synthetic scheme for intermediate 9.

**Appendix 1—chemical structure 2.** Chemical structures and synthetic scheme for avenolide.

## Synthesis of aldehyde fragment

For the synthesis of (R)−2-((4-methoxybenzyl)oxy)−2-methylbutanal (12), PPTS (46.85 mg, 0.185 mmol) and *p*-anisaldehyde dimethylacetal (0.63 ml, 3.73 mmol) at room temperature (RT) under $N_2$ were added to a solution of **1** (388 mg, 3.73 mmol) in $CH_2Cl_2$ (15 mL). After stirring for 5 hr at RT, the reaction was quenched with water and the aqueous phase was extracted with EtOAc (2 × 30 mL). The combined organic extracts were washed with brine and collected organic extracts were dried over anhydrous $Na_2SO_4$ and concentrated in vacuo using rotatory evaporator. The dried residue was purified by flash column chromatography (15% EtOAc/hexanes) to afford the corresponding *p*-methoxybenzylidene acetal (0.595 g, 92%) as a colorless oil.

1H-NMR (500 MHz, $CDCl_3$) δ 7.42 (d, *J* = 8.6 Hz, 2H), 6.90 (d, *J* = 8.6 Hz, 2H), 5.81 (s, 1H), 3.83 (s, 2H), 3.81 (s, 3H), 1.79–1.62 (m, 2H), 1.38 (s, 3H), 0.99 (t, *J* = 7.5 Hz, 3H); $^{13}$C-NMR (125 MHz, $CDCl_3$) δ 160.2, 130.5, 128.1, 113.6, 103.3, 81.7, 75.3, 55.2, 31.1, 24.2, 8.6; HRMS (ESI+, TFA-Na) calcd for $C_{13}H_{19}O_3$: 223.1334 [M+H]+, found m/z 223.1332.

DIBALH (1.02 M in hexanes, 4.00 mL, 4.05 mmol) at −78°C under $N_2$ was added to a solution of the *p*-methoxybenzylidene acetal (300 mg, 1.35 mmol) in $CH_2Cl_2$ (15 mL). After stirring for 1 hr at 0°C, the reaction was cautiously quenched with MeOH at 0°C, diluted with $CH_2Cl_2$ (30 mL) and treated with celite (1.20 g) and $Na_2SO_4.10H_2O$ (1.20 g). The mixture was allowed to warm to RT. After stirring at RT for 2 hr, the mixture was filtered through a celite pad, washed with DCM three times and the filtrate was concentrated in vacuo. The residue was purified by flash column chromatography (25% EtOAC/hexanes) to give the corresponding alcohol **2** (140 mg, 92%) as a colorless oil.

$^1$H-NMR (500 MHz, $CDCl_3$) δ 7.27 (d, *J* = 10 Hz, 2H), 6.89 (d, *J* = 10 Hz, 2H), 4.9 (s, 2H), 3.82 (s, 3H), 3.59 (d, *J* = 10 Hz, 1H), 3.55 (d, *J* = 10 Hz, 1H), 2.41 (brs, 1H), 1.66 (dq, *J* = 2.3 Hz, 7.4 Hz, 1H), 1.25 (s, 3H), 0.95 (t *J* = 7.5 Hz, 3H); $^{13}$C-NMR (125.0 MHz, $CDCl_3$) δ 159.1, 131.1, 129.1, 113.9, 77.9, 66.9, 63.4, 55.3, 27.8, 19.7, 8.1; HRMS(ESI) calcd for $C_{13}H_{19}O_3$; 223.1334 [M+H]+, found m/z 223.1338.

A solution of the oxalyl chloride (1.15 mL from 2 M stock in DCM, 2.23 mmol) in dry $CH_2Cl_2$ (7 mL) was cooled to −78°C under an atmosphere of Ar. A solution of dry DMSO (316 µL in 2 mL DCM, 4.46 mmol) was added slowly until temperature is maintained below −65°C. After stirring for 5 min, a solution of **2** (500 mg, 2.23 mmol) in dry $CH_2Cl_2$ (3 mL) was added slowly, and the resulting mixture was stirred slowly for 15 min, after which, $Et_3N$ (1.55 mL, 11.14 mmol) was added slowly. After stirring the reaction for 10 additional minutes at −78°C, the cooling bath was removed, and the reaction was allowed to warm to RT. Upon reaching RT, water (10 mL) was added and stirring continued for 15 min. The reaction mixture was then washed successively with 5% HCl (10 mL), saturated $NaHCO_3$ (10 mL) and brine (7 mL) using a separatory funnel. Collected organic layers were combined and dried with anhydrous $Na_2SO_4$, filtered and concentrated under reduced pressure to afford crude oil. Flash chromatography (20% EtOAc/hexanes) was performed to get pure aldehyde **12** (430 mg, 94%) as a yellow oil.

Spectroscopic characterization parameters agreed with the reported mass and chemical shift values (*Uchida et al., 2011*).

$^1$H-NMR (500 MHz, CDCl$_3$) δ 9.67 (s, 1H), 7.32 (d, *J* = 8.5 Hz, 2H), 6.91 (d, *J* = 10 Hz, 2H), 4.42 (d, *J* = 17.5 Hz,1H), 4.38 (d, *J* = 17.6 Hz, 1H), 3.84 (s, 3H), 1.84–1.67 (m, 2H) 1.34 (s, 3H), 0.95 (t, *J* = 7.5 Hz, 3H); $^{13}$C-NMR (500 MHz, CDCl$_3$) δ 205.3, 159.3, 129.1, 113.9, 82.8, 65.8, 55.3, 27.7, 17.7, 7.3; HRMS (ESI) calcd for C$_{13}$H$_{18}$NaO$_3$ 245.1154 [M+Na]+, found m/z 245.1152.

## Synthesis of alkene fragment

For synthesis of 5-(tert-Butyldiphenylsilanyloxy)-pentanal (**4**), 1,5-pentanediol (5 ml, 47.5 mmol) and TBDPSCl (12.35 mL, 47.5 mmol) were added to a solution of NaH (1.94 g, 47.5 mmol) in anhydrous THF (158 mL) at 0°C under N$_2$. The reaction mixture was stirred for 5 min at 0°C, then allowed to warm to RT. After stirring the reaction mixture for 2.5 hr at RT, the reaction was quenched with cold H$_2$O at 0°C, and the aqueous phase was extracted with CH$_2$Cl$_2$. The collected organic layer was washed with brine, and combined organic extracts were dried over anhydrous Na$_2$SO$_4$ and concentrated in vacuo. The residue was purified by flash column chromatography (15% EtOAc/hexanes) to afford the corresponding TBDPS ether, **3** (14 g, 86%) as a colorless oil (*Uchida et al., 2011*).

$^1$H-NMR (500 MHz, CDCl$_3$) δ 7.72 (d, 4H), 7.47–7.40 (m, 6H), 3.68 (t, *J* = 6.5 Hz, 2H), 3.68 (t, *J* = 6.5 Hz, 2H), 1.62–1.52 (m, 4H), 1.45–1.30 (m, 3H), 1.06 (s, 9H); $^{13}$C-NMR (500 MHz, CDCl$_3$) δ 135.8, 134.3,129.3, 127.9, 64.1, 63.2, 32.7, 32.5, 27.1, 22.2, 19.5; HRMS (ESI +, calcd for C$_{21}$H$_{30}$O$_2$Si 342.2015 [M+H]+, found m/z 342.2085.

A solution of dry DMSO (406 µL in 2 mL dry DCM, 5.721 mmol) was added slowly to a solution of the oxalyl chloride (1.43 mL from 2 M stock in DCM, 2.86 mmol) in dry CH$_2$Cl$_2$ (7 mL) cooled to −78°C under an atmosphere of Ar, to maintain temperature below −65°C. After stirring for 5 min, a solution of TBDPS ether **4** (980 mg, 2.86 mmol) in dry CH$_2$Cl$_2$ (3 mL) was added slowly, and the resulting mixture was stirred slowly for 15 min, after which, Et$_3$N (1.99 mL, 14.3 mmol) was added slowly. After stirring the reaction for 10 additional minutes at −78°C, the cooling bath was removed, and the reaction was allowed to warm to RT. Upon reaching RT, water (12 mL) was added and stirring continued for 15 min. The reaction mixture was then washed successively with 5% HCl (12 mL), saturated NaHCO$_3$ (12 mL) and brine (10 mL) using a separatory funnel. Collected organic layers were combined and dried with anhydrous Na$_2$SO$_4$, filtered and concentrated under reduced pressure to afford crude oil. Flash chromatography (20% EtOAc/hexanes) was performed to obtain pure aldehyde **4** (941 mg, 93%) as pale yellow oil.

$^1$H-NMR (500 MHz, CDCl$_3$) δ 7.72 (d, 4H), 7.47–7.40 (m, 6H), 3.68 (t, *J* = 6.5 Hz, 2H), 3.68 (t, *J* = 6.5 Hz, 2H), 1.62–1.52 (m, 4H), 1.45–1.30 (m, 3H), 1.06 (s, 9H); $^{13}$C-NMR (500 MHz, CDCl$_3$) δ 202.85, 135.8, 134.1, 129.9, 127.9, 63.6, 43.8, 32.1, 32.5, 27.1, 19.5, 18.9; HRMS (ESI +, calcd for C$_{21}$H$_{30}$O$_2$Si 340.1859 [M+H]+, found m/z 341.2015.

For synthesis of (E)−7-((tert-butyldiphenylsilyl)oxy)-hept-2-ene-1-ol (**8**), a solution of 5-((tert-butyldiphenylsilyl)oxy)-pentanal (3.5 g, 10.47 mmol) and (carboxymethylene) triphenylphosphorane (3.64 g, 10.47 mmol) in anhydrous CH$_2$Cl$_2$ (35 mL) was heated to reflux for 18 hr. The reaction was cooled to RT, diluted with water and extracted with pentane (2 × 200 mL). The collected organic extract was dried over anhydrous Na$_2$SO$_4$, filtered, and concentrated under reduced pressure to afford a crude yellow oil. Flash chromatography (15% EtOAc/hexanes) gave (E)−7-((tert-butyldiphenylsilyl)oxy)-hept-2-enoic acid ethyl ester (**5**) (3.14 g, 86%) as a yellow oily compound.

$^1$H-NMR (500 MHz, CDCl$_3$) δ 7.74 (dt, *J* = 9.3,6.3 Hz, 4H), 7.49–7.42 (m, 6H), 7.03 (dt, *J*= 15.7, 6.9 Hz, 1H), 5.89 (d, *J* = 15.6 Hz, 1H), 4.26 (q, *J*= 7.1 Hz, 2H), 3.72 (t, *J*= 5.6 Hz, 2H), 2.25 (q, *J*= 6.9 Hz, 2H), 1.68–1.61 (m, 4H), 1.36 (t, 7.1 Hz, 3H), 1.12 (s, 9H); $^{13}$C-NMR (500 MHz, CDCl$_3$) d 166.9, 149.4, 135.8, 134.2, 129.8, 121.6, 63.7, 60.3, 32.1, 27.1, 24.6, 19.5, 14.6; HRMS (ESI+, TFA-Na) calcd for C$_{25}$H$_{34}$NaO$_3$Si 433.2175. [M+Na]+, found m/z 433.2161.

To a solution of the corresponding α,β-unsaturated ester(**5**) (5.9 g, 14.87 mmol) in anhydrous CH$_2$Cl$_2$ (100 mL) was added DIBALH (1.02 M in hexanes, 30.6 mL, 31.2 mmol) at

−78°C under N$_2$. The mixture was then warmed to 0°C. After stirring for 1 hr at 0°C, the reaction was cautiously quenched with cold MeOH at 0°C, diluted with CH$_2$Cl$_2$ and celite (14.2 g) and Na$_2$SO$_4$.10H$_2$O (14.2 g) were added to it. The mixture was allowed to warm to RT and stirred for 2 hr, then filtered through a pad of celite and the filtrate was concentrated under reduced pressure. The residue was purified by flash column chromatography (30% EtOAc/hexanes) to give alcohol 6 (5.5 g, quant.) as a colorless oil.

$^1$H-NMR (500 MHz, CDCl$_3$) δ 7.71 (dt, J = 1.4,6.5 Hz, 4H), 7.49–7.37 (m, 6H), 5.69 (m, 2H), 4.12 (d, J= 5.6 Hz, 2H), 3.70 (t, J= 6.4 Hz, 2H), 2.07 (q, J= 7.1 Hz, 2H), 1.63–1.46 (m, 4H), 1.09 (s, 9H); $^{13}$C-NMR (500 MHz, CDCl$_3$) δ 135.8, 134.3, 133.4, 129.8, 129.3, 127.9, 64.05, 32.2, 27.2, 25.6, 19.5; HRMS (ESI+, TFA-Na) calcd for C$_{23}$H$_{32}$NaO$_2$Si 391.2069. [M+Na]+, found m/z 391.2134.

PPTS (214.76 mg, 0.854 mmol) and DHP (7.8 mL, 85.4 mmol) were added to a solution of allyl alcohol 7 (3.15 g, 8.54 mmol) in anhydrous CH$_2$Cl$_2$ (40 mL) at 0°C under N$_2$. The reaction mixture was then stirred for 6 hr at 0°C followed by quenching with H$_2$O. The aqueous phase was extracted with EtOAc and combined organic extracts were washed with brine and dried over anhydrous Na$_2$SO$_4$ and concentrated in vacuo. The residue was purified by flash column chromatography (15% EtOAc/hexanes) to afford the corresponding THP ether(7) (3.5 g, **88%** yield) as a colorless oil (*Uchida et al., 2011*).

$^1$H-NMR (500 MHz, CDCl$_3$) δ 7.71 (d, J = 6.5, 4H), 7.47–7.41 (m, 6H), 5.72–5.69 (m, 1H), 5.63–5.57 (m, 1H), 4.68 (m, 1H), 4.24–4.21 (m, 1H), 3.99–3.90 (m, 2H), 3.70 (t, J = 6.2 Hz, 2H), 3.55 (m, 1H), 2.09 (m, 2H), 1.90 (m, 1H), 1.77 (m, 1H), 1.63 (m, 8H), 1.51 (m, 2H), 1.31 (m, 1H), 1.09 (s, 9H); $^{13}$C-NMR (500 MHz, CDCl$_3$) δ 135.8, 134.7, 134.3, 129.8, 126.5, 98.1, 68.1, 63.1, 64.0, 62.5, 32.3, 30.9, 27.1, 25.6; HRMS (ESI+, TFA-Na) calcd for C$_{28}$H$_{40}$NaO$_3$Si 475.2644 [M+Na]+, found m/z 475.2650.

TBAF (9.2 mL from 1 M solution in THF, 9.2 mmol) was added to a solution of the THP ether (7) (3.5 g, 7.74 mmol) in dry THF (55 mL) at RT under N$_2$. The reaction mixture turned orangish red in color and it was stirred for 4.5 hr at RT. The reaction was quenched with H$_2$O, and the aqueous phase was extracted with CH$_2$Cl$_2$. The combined organic extracts were washed with brine, and then dried over anhydrous Na$_2$SO$_4$ and concentrated in vacuo. The obtained yellowish-orange crude residue was purified by flash column chromatography (50% EtOAc/hexanes) to give the corresponding TBDPS deprotected alcohol (8) (1.45 g, 92%) as a colorless oil. The spectroscopic data match with the reported data (*Uchida et al., 2011*).

$^1$H-NMR (500 MHz, CDCl$_3$) δ 5.74–5.67 (m, 1H), 5.62–5.57 (m, 1H), 4.63–4.60 (m, 1H), 4.18 (ddd, J = 12.0, 5.7, 1.0 Hz, 1H), 3.93 (ddd, J = 12.0, 5.7, 1.0 Hz, 1H), 3.89–3.82 (m, 1H), 3.64 (t, J = 6.5 Hz, 2H), 3.52–3.48 (m, 1H), 2.12–2.04 (m, 2H), 1.86–1.43 (m, 10H); $^{13}$C-NMR (500 MHz, CDCl$_3$) δ 134.5, 126.5, 97.9, 68.0, 62.8, 62.4, 32.4, 32.2, 30.8, 25.6, 25.4, 19.7; HRMS (FAB, m-NBA) calcd for C$_{12}$H$_{22}$NaO$_3$ 273.1467 [M+Na]+, found m/z 273.1469.

For the synthesis of (E)−2-((7-Iodohept-2-en-1-yl)oxy)tetrahydro-2H-pyran (11), Et$_3$N (1.3 mL, 9.33 mmol), Me$_3$N.HCl (44.5 mg, 0.47 mmol) and MsCl (541 μl, 6.99 mmol) were added to a solution of 8 (1 g, 4.66 mmol) in CH$_2$Cl$_2$ (50 mL)at 0°C under N$_2$ atmosphere. The reaction mixture was allowed to warm to RT and stirred for 3 hr at RT. After that, the reaction was quenched with H$_2$O and the aqueous phase was extracted with CH$_2$Cl$_2$. The combined organic extracts were dried over anhydrous Na$_2$SO$_4$ and concentrated under reduced pressure. The obtained yellowish-orange residue was purified by flash column chromatography (40% EtOAc/hexanes) to afford the corresponding mesylate ((E)−7-((tetrahydro-2H-pyran-2-yl)oxy) hept-5-en-1-yl methanesulfonate)(9) (1.35 g, quant.) as a colorless oil. The spectroscopic data match with the reported data (*Uchida et al., 2011*).

$^1$H-NMR (500MHz, CDCl$_3$) δ 5.69–5.64 (m, 1H), 5.62–5.55 (m, 1H), 4.61–4.60 (m, 1H), 4.21 (t, J = 6.5 Hz, 2H), 4.17 (ddd, J = 12.0, 5.3, 1.2 Hz, 1H), 3.91 (ddd, J = 12.0, 5.3, 1.2 Hz, 1H), 3.88–3.47 (m, 1H), 3.51–3.47 (m, 1H), 2.99 (s, 3H), 2.12–2.06 (m, 2H), 1.84–1.46 (m, 10H); $^{13}$C-NMR (500MHz, CDCl$_3$) δ 133.0, 127.0, 97.8, 69.8, 67.6, 62.2, 37.2, 31.5, 30.5, 28.5, 25.3, 24.7, 19.4; HRMS (ESI+, TFA-Na) calcd for C$_{13}$H$_{24}$NaO$_5$S 315.1242 [M+Na], found m/z 315.1242.

Following the protocol reported by *Uchida et al., 2011*, the OH was substituted with an iodo group. Briefly, the mesyl ester (9) (3.00 g, 10.46 mmol) was added to a solution of

NaCl (2.30 g, 15.4 mmol) in acetone (150 mL) under $N_2$ atmosphere. After stirring at reflux for 8.5 hr, the reaction mixture was cooled to RT and the aqueous phase was extracted with EtOAc. Organic fractions were combined, dried over anhydrous $Na_2SO_4$ and concentrated in vacuo. The residue was purified by flash column chromatography (10% EtOAc/hexanes) to give the corresponding iodo compound (**11**) (2.0 g, 62%) as a yellow colored oil. The spectroscopic data match with the reported data (*Uchida et al., 2011*).

[1]H-NMR (500MHz, CDCl$_3$) δ 5.75–5.58 (m, 2H), 4.67 (dd, J = 4.3, 2.9 Hz, 1H), 4.21 (ddd, J = 12.0, 5.4, 1.0 Hz, 1H), 3.93 (ddd, J = 12.0, 5.3, 1.2 Hz, 1H), 3.92–3.88 (m, 1H), 3.57–3.52 (m, 1H), 3.23 (t, J = 7.0 Hz, 2H), 2.14–2.10 (m, 2H), 1.90–1.51 (m, 10H); [13]C-NMR (500MHz, CDCl$_3$) δ 133.7, 127.1, 98.0, 67.9, 62.5, 33.2, 31.4, 30.9, 30.1, 25.7, 19.8, 7.0; HRMS (FAB, m-NBA): calcd for $C_{12}H_{21}NaO_2I$ 347.0484 [M+Na]+, found m/z 347.0480.

Following protocol reported by *Uchida et al., 2011*, allyl alcohol (**16**) was synthesized from intermediates **11** and **12** via intermediates **14** (diastereoisomeric mixture) and ketone (**15**).

For the synthesis of (R)−8-((2S,3S)−3-(hydroxymethyl)oxiran-2-yl)−3-((4-methoxybenzyl)oxy)−3-methyloctan-4-one (**17**), Ti(OiPr)$_4$ (728 µl, 2.4 mmol) was added to a suspension of activated 4 Å molecular sieves (320 mg) and (+)-DET (422 µl, 2.4 mmol) in dry $CH_2Cl_2$ (15.0 mL) at −20°C under an atmosphere of Ar. After stirring for 0.5 hr, t-BuOOH (985.6 µl, 4.8 mmol) was slowly added to the suspension at −20°C and the resulting mixture was continued to stir at −20°C for 0.5 hr. A solution of **16** (800 mg, 2.4 mmol) in $CH_2Cl_2$ (20 mL) was then added dropwise to the reaction mixture at a constant rate, and the mixture was stirred for 18 hr at −20°C. The reaction was quenched with Me$_2$S (250 µl, 3.35 mmol) in 2 mL DCM, diluted with $CH_2Cl_2$ (20 mL) and treated with celite (2.62 g) and $Na_2SO_4.10H_2O$ (2.62 g). The suspension was allowed to warm to RT and then stirred for 2 hr. The resulting mixture was filtered through a pad of celite and the filtrate was concentrated under reduced pressure. The residue was purified by flash column chromatography (50% EtOAc/hexanes) to afford **10** (311 mg, 86%) as a colorless oil. The spectroscopic parameters match exactly with the reported values (*Uchida et al., 2011*).

[1]H-NMR (500MHz, CDCl$_3$) δ 7.29 (d, J = 8.8 Hz, 2H), 6.91 (d, J = 8.8 Hz, 2H), 4.36 (d, J = 10.7 Hz, 1H), 4.32 (d, J = 10.7 Hz, 1H), 3.92–3.85 (m, 1H), 3.84 (s, 3H), 3.67–3.64 (m, 1H), 2.96–2.88 (m, 2H), 2.69 (dt, J = 7.2, 2.6 Hz, 2H), 1.89–1.77 (m, 2H), 1.75–1.58 (m, 4H), 1.50–1.41 (m, 2H), 1.35 (s, 3H), 0.87 (t, J = 7.5 Hz, 3H); [13]C-NMR (500MHz, CDCl$_3$) δ 214.9, 159.0, 130.7, 128.6, 113.8, 84.9, 65.1, 61.7, 58.4, 55.8, 55.3, 36.8, 31.4, 29.2, 25.7, 23.1, 20.0, 7.91; HRMS (ESI+, TFA-Na) calcd for $C_{20}H_{30}NaO_5$ 373.1991 [M+Na]+, found m/z 373.1989.

For the synthesis of (3S,9R)−9-((4-methoxybenzyl)oxy)−9-methyl-8-oxoundec-1-en-3-yl acrylate (**19**), acryloyl chloride (87.1 µl, 1.06 mmol), Et$_3$N (300 µl, 2.13 mmol) and DMAP (8.5 mg, 0.07 mmol, cat.) were added to a solution of the allyl alcohol (**17**) (230 mg, 0.71 mmol) in $CH_2Cl_2$ (10 mL) under $N_2$. The mixture was warmed to RT and stirred for 1 hr at RT. After that, the reaction was quenched with $H_2O$ and the aqueous phase was extracted three times with $CH_2Cl_2$. Organic layers were combined and dried over anhydrous $Na_2SO_4$ and concentrated in vacuo. The residue was purified by flash column chromatography (80% hexanes/EtOAc) to afford acrylyl alkene (**18**) (220 mg, 90%) as a colorless oil. The spectroscopic data match with the reported values (*Uchida et al., 2011*).

[1]H-NMR (500 MHz, CDCl$_3$) δ 7.31 (d, J = 8.8 Hz, 2H), 6.93 (d, J = 8.8 Hz, 2H), 6.45 (dd J = 17.3, 1.5 Hz, 1H), 6.16 (dd, J = 17.3, 10.4 Hz, 1H), 5.86–5.76 (m, 2H), 5.33–5.29 (m, 1H), 5.25 (ddd, J = 17.3, 9.3, 1.3 Hz, 1H), 5.20 (ddd, J = 10.5, 5.9, 1.2 Hz, 1H), 4.36 (d, J = 10.7 Hz, 1H), 4.31 (d, J = 10.7 Hz, 1H), 3.84 (s, 3H), 2.68 (dt, J = 7.3, 3.9 Hz, 2H), 1.89–1.55 (m, 8H), 1.35 (s, 3H), 0.87 (t, J = 7.5 Hz, 3H); [13]C-NMR (500 MHz, CDCl$_3$) δ 215.0, 165.7, 159.3, 136.5, 130.9, 130.9, 128.9, 128.9, 117.0, 114.0, 113.8, 85.1, 75.0, 65.3, 55.5, 37.0, 34.3, 29.4, 25.0, 23.5, 20.2, 8.1; HRMS (ESI+, TFA-Na) calcd for $C_{23}H_{32}NaO_5$ 411.2147 [M+Na]+, found m/z 411.2136.

162 mg of DDQ (0.69 mmol) was added to a solution of **18** (208 mg, 0.53 mmol) in $CH_2Cl_2$:$H_2O$ (20:1, 5.2 mL total) at RT under $N_2$. After stirring for 1 hr at RT, the reaction was quenched with $H_2O$ and the aqueous phase was extracted with $CH_2Cl_2$. The combined organic extracts were dried over anhydrous $Na_2SO_4$ and concentrated in vacuo. The residue was purified by flash column chromatography (20–25% EtOAc/hexanes) to give the corresponding

alcohol (**19**) (125 mg, quant.) as a colorless oil. All spectroscopic and optical density data matched exactly with the reported values (*Uchida et al., 2011*).

For the synthesis of (4S,10R)-avenolide (**13**), a solution of Grubbs second-generation catalyst (15 mg, 0.026 mmol) in $CH_2Cl_2$ (23 mL) was added to a solution of the alcohol **19** (100 mg, 0.37 mmol) in $CH_2Cl_2$ (12.5 mL) at RT under $N_2$. After 2 hr of stirring at 40°C, Quadrasil AP (1.25 g) was added to the reaction mixture. The suspension was stirred for an additional 5 min at RT and then allowed to stand for 10 min. The mixture was filtered through a pad of celite and the filtrate was washed with $H_2O$. The organic layers were dried over anhydrous $Na_2SO_4$ and concentrated in vacuo. The residue was purified twice by flash column chromatography (50% EtOAC/hexanes) to afford **13** (28 mg, 78%) as a light brownish oil. All optical density and spectroscopic values matched the reported values (*Uchida et al., 2011*).

$[\alpha]^{26}_D$ (c 1.0, $CHCl_3$): +2.24; $^1$H-NMR (500MHz, $CDCl_3$) δ 7.49 (dd, J = 5.8, 1.5 Hz, 1H), 6.17 (dd, J = 5.7, 2.0 Hz, 1H), 5.10–5.03 (m, 1H), 3.80 (brs, 1H), 2.59–2.47 (m, 2H), 1.87–1.61 (m,6H), 1.61–1.42 (m, 2H), 1.38 (s, 3H), 0.86 (t, J = 7.4 Hz, 3H); $^{13}$C-NMR (500MHz, $CDCl_3$) δ 214.4, 173.2, 156.2, 121.9, 83.2, 79.1, 35.6, 33.2, 32.6, 25.4, 24.9, 23.4, 7.9; HRMS (ESI+) calcd for $C_{13}H_{21}O_4$ 241.1440 [M+H]+, found m/z 241.1441.

# $^1$H and $^{13}$C NMR Spectra

**Appendix 1—figure 1.** Proposed biosynthetic pathway for avenolide and putative roles of biosynthetic genes.

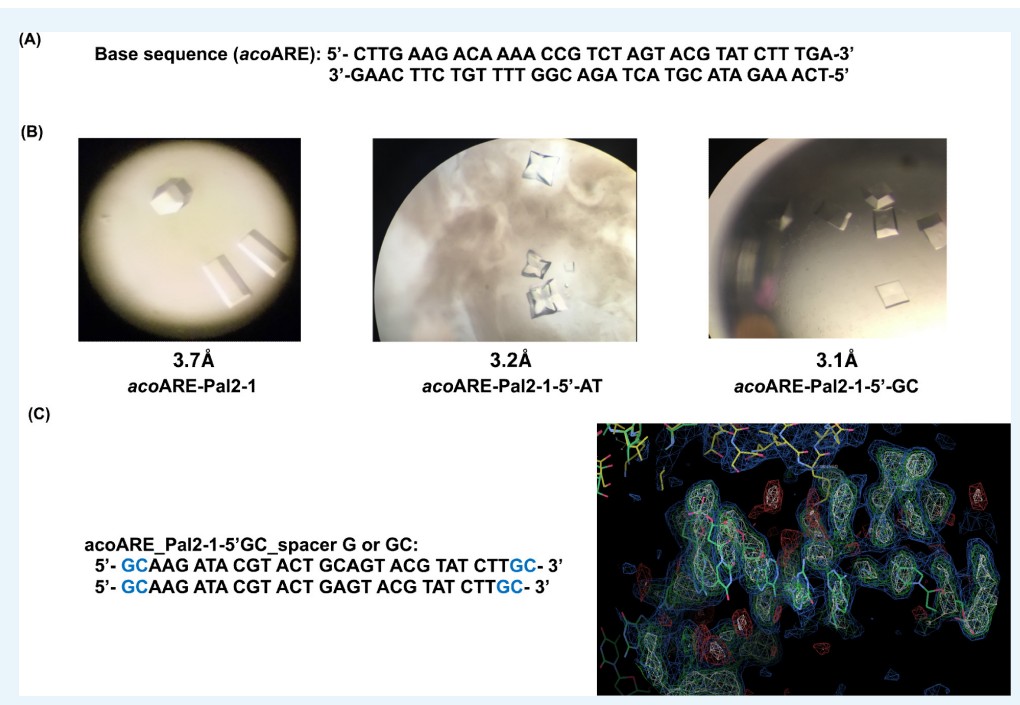

**(A)**

Base sequence (*aco*ARE): 5'- CTTG AAG ACA AAA CCG TCT AGT ACG TAT CTT TGA-3'
3'-GAAC TTC TGT TTT GGC AGA TCA TGC ATA GAA ACT-5'

**(B)**

| 3.7Å | 3.2Å | 3.1Å |
|------|------|------|
| *aco*ARE-Pal2-1 | *aco*ARE-Pal2-1-5'-AT | *aco*ARE-Pal2-1-5'-GC |

**(C)**

acoARE_Pal2-1-5'GC_spacer G or GC:
5'- GCAAG ATA CGT ACT GCAGT ACG TAT CTTGC- 3'
5'- GCAAG ATA CGT ACT GAGT ACG TAT CTTGC- 3'

**Appendix 1—figure 2.** Crystallographic studies with DNA oligonucleotides. (**a**) AvaR1 binding site in the upstream region of *aco* gene. (**b**) Crystals obtained with DNA Oligo Pal2-1 variant sequences; below the structure on the right is the screenshot from COOT depicting the electron density of the DNA oligo. (**c**) Proposed sequences containing linker with 1 bp or 2bp length.

## Appendix tables

**Appendix 1—table 1. List of all of the *aco*ARE oligonucleotides tried for co-crystallization with AvaR1.** Pal in the name denotes the palindromic sequences that have been designed using the first or the second half of the symmetric sequence, which are self-annealing. The first half of the sequence is complementary to the second half.

Target sequence (*aco*ARE): 5'-CTTGAAGACAAAACCGTCTAGTACGTATCTTTGA-3' 3'-GAAC TTC TGTTTTGGCAGATCATGCATAGAAACT- 5'

| Aco_ARE_Oligo | Sequence |
|---------------|----------|
| *aco*ARE +1 | 5'-GAAGACAAAACCGTCTAGTACGTATCTTTGA-3'<br>3'-CTTC TGTTTTGGCAGATCATGCATAGAAACT-5' |
| *aco*ARE +4 | 5'-CTTGAAGACAAAACCGTCTAGTACGTATCTTTGACCT-3'<br>3'-GAACTTC TGTTTTGGCAGATCATGCATAGAAACTGGA-5' |
| *aco*ARE +5 | 5'-ACTTGAAGACAAAACCGTCTAGTACGTATCTTTG ACCTC-3'<br>3'-TGAACTTC TGTTTTGGCAGATCATGCATAGAAACTGGAG-5' |
| *aco*ARE_pal1 | 5'-TTG AAG ACA AAA CCG TCT AGA CGG TTT TGT CTT CAA-3' |
| *aco*ARE_pal2 | 5'-TCA AAG ATA CGT ACT AGT ACG TAT CTT TGA- 3' |
| *aco*ARE_pal1 +one each | 5'-CTTG AAG ACA AAA CCG TCT AGA CGG TTT TGTCTTCAAG-3' |
| *aco*ARE_pal2 +one each | 5'-GTCA AAG ATA CGT ACT AGT ACG TAT CTT TGAC-3' |
| *aco*ARE_pal1-5' over A | 5'-ATTG AAG ACA AAA CCG TCT AGA CGG TTT TGT CTT CAA-3' |
| *aco*ARE_pal2 5' over A | 5'-ATCA AAG ATA CGT ACT AGT ACG TAT CTT TGA-3' |
| *aco*ARE_pal1-2 each | 5'- G AAG ACA AAA CCG TCT AGA CGG TTT TGT CTT C-3' |
| *aco*ARE_pal2-1 each | 5'- CA AAG ATA CGT ACT AGT ACG TAT CTT TG- 3' |

*Appendix 1—table 1 continued on next page*

*Appendix 1—table 1 continued*

| Aco_ARE_Oligo | Sequence |
|---|---|
| *aco*ARE_pal2-2 each | 5'- A AAG ATA CGT ACT AGT ACG TAT CTT T- 3' |
| *aco*ARE_pal2-1-3'G | 5'- CA AAG ATA CGT ACT AGT ACG TAT CTT T- 3' |
| *aco*ARE_pal2-1each-5'C | 5'- CCA AAG ATA CGT ACT AGT ACG TAT CTT TG- 3' |
| *aco*ARE_pal2-1each-3'G | 5'- CA AAG ATA CGT ACT AGT ACG TAT CTT TGG- 3' |
| acoARE_pal2 −1e+GCpair | 5'- GCA AAG ATA CGT ACT AGT ACG TAT CTT TGC- 3' |
| acoARE_Pal2-3each | 5'-AAG ATA CGT ACT AGT ACG TAT CTT- 3' |
| *aco*ARE_Pal2-1-5'CG | 5'-CGAAG ATA CGT ACT AGT ACG TAT CTT CG- 3' |
| *aco*ARE_Pal2-1-5'GC | 5'- GCAAG ATA CGT ACT AGT ACG TAT CTT GC- 3' |
| *aco*ARE_Pal2-1-5'TA | 5'- TAAAG ATA CGT ACT AGT ACG TAT CTT TA- 3' |
| *aco*ARE_Pal2-1-5'AT | 5'-ATAAG ATA CGT ACT AGT ACG TAT CTT AT- 3' |
| *aco*ARE_Pal2-1-5'GC-Mid G | 5'- GCAAG ATA CGT ACTG AGT ACG TAT CTT GC- 3' |
| *aco*ARE_Pal2-1-5'GC-Mid GC | 5'- GCAAG ATA CGT ACTGC AGT ACG TAT CTT GC- 3' |

**Appendix 1—table 2. List of identified *Streptomyces* strains with homology to *aco*, *avar1*, and *cyp* genes involved in avenolide biosynthesis in *S. avermitilis*.** Strains are from the genus *Streptomyces* unless otherwise noted.

| Receptor | Aco | Cyp450 | Strain |
|---|---|---|---|
| ADK59_29015 | ADK59_RS28945 | ADK59_RS28935 | XY332 |
| ADK54_RS22575 | ADK54_RS22580 | ADK54_RS22570 | WM6378 |
| SPRI_RS01555 | SPRI_RS01560 | SPRI_RS01550 | *S. pristinaespiralis* |
| SPRI_RS34865/spbR | SPRI_RS34860 | SPRI_RS34870 | *S. pristinaespiralis* |
| AVL59_26260 | AVL59_RS26265 | AVL59_RS26255 | *S. griseochromogenes* strain ATCC 14511 |
| ASE41_15570/scaR | ASE41_RS08690 | ASE41_RS08700 | *Streptomyces* sp. Root264 |
| B446_03460 | B446_RS03395 | B446_RS03385 | *S. collinus* (strain DSM 40733/Tu 365) |
| SAZU_2710 | AOQ53_RS12925 | AOQ53_RS12915 | *S. azureus* strain ATCC 14921 |
| tylP | orf18 | orf16 | *S. fradiae* |
| TU94_00975 | TU94_RS00980 | TU94_RS00965 | *S. cyaneogriseus* subsp. noncyanogenus |
| AT728_21175 | AT728_RS06415 | AT728_RS06425 | *S. silvensis* |
| SSFG_07848 | SSFG_07849 | SSFG_07847 | *S. viridosporus* ATCC 14672 |
| AQJ91_00095 | AQJ91_RS00090 | AQJ91_RS00100 | RV15 |
| SGLAU_25540 | SGLAU_RS25200 | SGLAU_RS25210 | *S. glaucescens* |
| AQI88_17505 | AQI88_RS17500 | AQI88_RS17510 | *S. cellostaticus* |
| avaR1 | aco/SM007_06205 | cyp17 | *S. avermitilis* |
| BEN35_RS25960 | BEN35_RS25965 | BEN35_RS25955 | *S. fradiae* strain Olg4R |
| SGM_6044 | SGM_6045 | SGM_6043 | *S. griseoaurantiacus* M045 |
| AQJ66_RS29075 | AQJ66_29065 | AQJ66_RS29080 | *S. bungoensis* |
| BIV23_RS09990 | BIV23_RS09995 | BIV23_RS09985 | MUSC 1 |
| OP17_RS26145 | OP17_RS26140 | OP17_RS26150 | *S. aureofaciens* strain NRRL B-1286 |
| AOK23_RS06340 | AOK23_RS06335 | AOK23_RS06345 | *S. torulosus* strain NRRL B-3889 |

*Appendix 1—table 2 continued on next page*

*Appendix 1—table 2 continued*

| Receptor | Aco | Cyp450 | Strain |
|---|---|---|---|
| AOK12_RS18690 | AOK12_RS18695 | AOK12_RS18685 | *S. kanamyceticus* strain NRRL B-2535 |
| AOK14_RS28840 | AOK14_RS28845 | AOK14_RS28835 | *S. neyagawaensis* strain NRRL B-3092 |
| JHAT_RS31450 | JHAT_RS31455 | JHAT_RS31445 | JHA26 |
| IG92_RS0101750 | IG92_RS0101755 | IG92_RS0101745 | *S. cacaoi* subsp. *cacaoi* NRRL S-1868 |
| IH57_RS0113175 | IH57_RS0113170 | IH57_RS0113180 | NRRL F-5053 |
| TR46_RS36115 | TR46_RS36110 | TR46_RS36120 | *Streptacidiphilus carbonis* strain NBRC 100919 |
| AWZ10_RS30605 | AWZ10_RS30600 | AWZ10_RS30610 | *S. europaeiscabiei* strain 96–14 |
| AMK31_RS05975 | AMK31_RS05980 | AMK31_RS05970 | TSRI0107 |
| OQI_RS18015 | OQI_RS18020 | OQI_RS18010 | *S. pharetrae* CZA14 |
| AOK15_RS33540 | AOK15_RS33545 | AOK15_RS33535 | *S. ossamyceticus* strain NRRL B-3822 |
| AOK17_RS00790 | AOK17_RS00795 | AOK17_RS00785 | *S. phaeochromogenes* strain NRRL B-1248 |
| B079_RS0125750 | B079_RS0125745 | B079_RS0125755 | LaPpAH-108 |
| AMK33_RS39295/ AMK33_38290 | AMK33_RS39290 | AMK33_RS39300 | CB02400 |
| ASC56_RS07050 | ASC56_RS07055 | ASC56_RS07045 | TP-A0356 |
| BEK98_43205 | BEK98_43200 | BEK98_RS44190 | *S. diastatochromogenes* |
| SAMN04487983_101174 | SAMN04487983_101173 | SAMN04487983_101175 | yr375 |
| B5181_21375 | B5181_21380 | B5181_21370 | 4F |
| B9W62_10200 | B9W62_10205 | B9W62_10195 | CS113 |
| SAMN05216260_11022 | SAMN05216260_11023 | SAMN05216260_11021 | *S. jietaisiensis* |
| BN2145_RS03090 | BN2145_RS03095 | BN2145_RS03085 | *S. leeuwenhoekii* |
| KY5_6076 | KY5_6075 | KY5_6077 | *S. formicae* |
| CW362_40715/ CW362_RS40740 | CW362_40710 | CW362_40720 | *S. populi* |
| SAMN05421806_12721 | SAMN05421806_12722 | SAMN05421806_12720 | *S. indicus* |
| CTU88_08915 | CTU88_08920 | CTU88_08910 | JV178 |
| BX282_0700 | BX282_0701 | BX282_0699 | 1121.2 |
| SAMN06272765_6800 | SAMN06272765_6799 | SAMN06272765_6801 | Ag109_G2-15 |
| CJD44_11095 | CJD44_11100 | CJD44_11090 | alain-838 |
| C3488_RS02995 | C3488_RS03000 | C3488_RS02990 | Ru72 |
| C6Y14_RS06395 | C6Y14_06390 | C6Y14_RS06400 | A217 |
| IF73_RS0131080 | IF73_RS0131075 | IF73_RS0131085 | NRRL F-5727 |
| C6N75_16870/ C6N75_RS16880 | C6N75_16875 | C6N75_RS16865 | ST5x |
| VO63_07870 | VO63_07865 | VO63_07875 | *S. showdoensis* |
| IF54_RS0133395 | IF54_RS0133390 | IF54_RS0133400 | NRRL B-3229 |
| EW58_RS46355 | EW58_RS46360 | EW58_RS46350 | *S. mirabilis* |
| BG482_RS07255 | BG482_RS07260 | BG482_RS07250 | LUP30 |
| STEPF7_RS00065 | STEPF7_RS00060 | STEPF7_RS00070 | F-7 |
| C6376_26350 | C6376_26345 | C6376_26355 | P3 |

*Appendix 1—table 2 continued*

| Receptor | Aco | Cyp450 | Strain |
|---|---|---|---|
| BS75_RS38740 | BS75_RS38735 | BS75_RS38745 | *Streptacidiphilus albus* JL83 |
| SMA5143A_3910 | SMA5143A_3909 | SMA5143A_3911 | MA5143a |
| SLUN_38640 | SLUN_38645 | SLUN_38635 | *S. lunaelactis* |
| CLW08_6960/ CLW08_RS34500 | CLW08_6959 | CLW08_6961 | 69 |
| CLW15_0573 | CLW15_0574 | CLW15_0572 | 73 |
| DC095_032510 | DC095_032505 | DC095_032515 | *S. xinghaiensis* |
| C8R36_7975 | C8R36_7974 | C8R36_7976 | 3212.5 |
| CLW07_7979 | CLW07_7978 | CLW07_7980 | 67 |
| BX279_8804 | BX279_8803 | BX279_8805 | Ag82_O1-9 |
| C8R37_8029 | C8R37_8028 | C8R37_8030 | 3212.4 |
| DT_019_27550 | DT_019_27545 | DT_019_27555 | SDr-06 |
| EDD87_5077 | EDD87_5076 | EDD87_5078 | *S. ossamyceticus* |
| C4J65_35580 | C4J65_35585 | C4J65_35575 | CB09001 |
| DI272_14555 | DI272_14560 | DI272_14550 | Act143 |
| DKG34_25265 | DKG34_25260 | DKG34_25270 | NWU49 |
| CLW14_9027 | CLW14_9026 | CLW14_9028 | 75 |
| EDE03_1306 | EDE03_1307 | EDE03_1305 | Ag82_G5-5 |
| E2B92_31970 | E2B92_31965 | E2B92_31975 | WAC05374 |
| FE633_RS10505 | FE633_RS10500 | FE633_RS10510 | NEAU-C151 |
| FGD71_RS00515 | FGD71_RS00510 | FGD71_RS00520 | NEAU-SSA 1 |
| FNX44_RS06285 | FNX44_RS06290 | FNX44_RS06280 | OF1 |
| E4K73_RS21075 | E4K73_RS21070 | E4K73_RS21080 | IB201691-2A2 |
| EV585_RS00830 | EV585_RS00825 | EV585_RS00835 | BK335 |
| EV288_RS22310 | EV288_RS22315 | EV288_RS22305 | BK215 |
| DN402_06475 | DN402_06480 | DN402_06470 | SW4 |
| F3T56_RS11975 | F3T56_RS11980 | F3T56_RS11970 | TRM68348 |
| EV298_RS42585 | EV298_RS42590 | EV298_RS42580 | BK042 |
| ESG85_RS18290 | ESG85_RS18295 | ESG85_RS18285 | TRM44457 |
| EV588_RS28855 | EV588_RS28860 | EV588_RS28850 | BK141 |
| B9W62_10200 | B9W62_10205 | B9W62_10195 | CS113 |
| FB157_RS34410 | FB157_RS34405 | FB157_RS34415 | BK340 |
| TNCT1_RS20710 | TNCT1_RS20705 | TNCT1_RS20715 | 1–11 |
| Sri02f_RS33870 | Sri02f_RS33865 | Sri02f_RS33875 | *S. rishiriensis* strain NBRC 13407 |

**Appendix 1—table 3. Crystallographic refinement parameters.**

| | SeMet AvaR1 | AvaR1-avenolide | AvaR1-DNA |
|---|---|---|---|
| **Data collection** | | | |
| Space group | $P2_1$ | $P2_1$ | $P4_2$ |
| Cell: a, b, c (Å)/β (°) | 42.0, 78.9, 130.2/93.3 | 44.4, 232.5, 87.7/92.7 | 130.5, 130.5, 180.6 |
| Resolution (Å)[*] | 50–2.4 (2.5–2.40) | 116–2.0 (2.0–1.99) | 130–3.08 (3.13–3.08) |

*Appendix 1—table 3 continued on next page*

*Appendix 1—table 3 continued*

|  | SeMet AvaR1 | AvaR1-avenolide | AvaR1-DNA |
|---|---|---|---|
| **Data collection** | | | |
| Total reflections | 166,803 | 570,056 | 836,259 |
| Completeness (%) | 99.9 (98.9) | 95.9 (65.4) | 100 (100) |
| $R_{sym}$ (%) | 8.8 (62.8) | 13.4 (62.1) | 7.7 (127.5) |
| Redundancy | 5.2 (5.2) | 4.8 (5.5) | 15.1 (15.2) |
| $I/\sigma(I)$ | 7.6 (1.8) | 10.5 (2.5) | 25.9 (2.2) |
| Refinement | | | |
| Resolution (Å) | 39.3–2.4 | 25.0–2.0 | 25.0–3.09 |
| Number reflections | 31,643 | 110,860 | 52,651 |
| $R_{work}/R_{free}^{\dagger}$ | 22.3/27.5 | 19.9/24.1 | 19.5/26.3 |
| Number of atoms | | | |
| Protein | 6843 | 13,313 | 13,169 |
| Water | 249 | 1299 | – |
| DNA/ligand | – | 136 | 1148 |
| B-factors | | | |
| Protein | 54.3 | 22.9 | 105.3 |
| Water | 44.6 | 31.7 | – |
| DNA/ligand | – | 15.8 | 82.3 |
| R.M.S. deviations | | | |
| Bond lengths (Å) | 0.012 | 0.009 | 0.010 |
| Bond Angles (°) | 1.58 | 1.48 | 1.69 |

*Highest resolution shell is shown in parenthesis.

†R-factor = $\Sigma(|Fobs|-k|Fcalc|)/\Sigma|Fobs|$ and R-free is the R value for a test set of reflections consisting of a random 5% of the diffraction data not used in refinement.

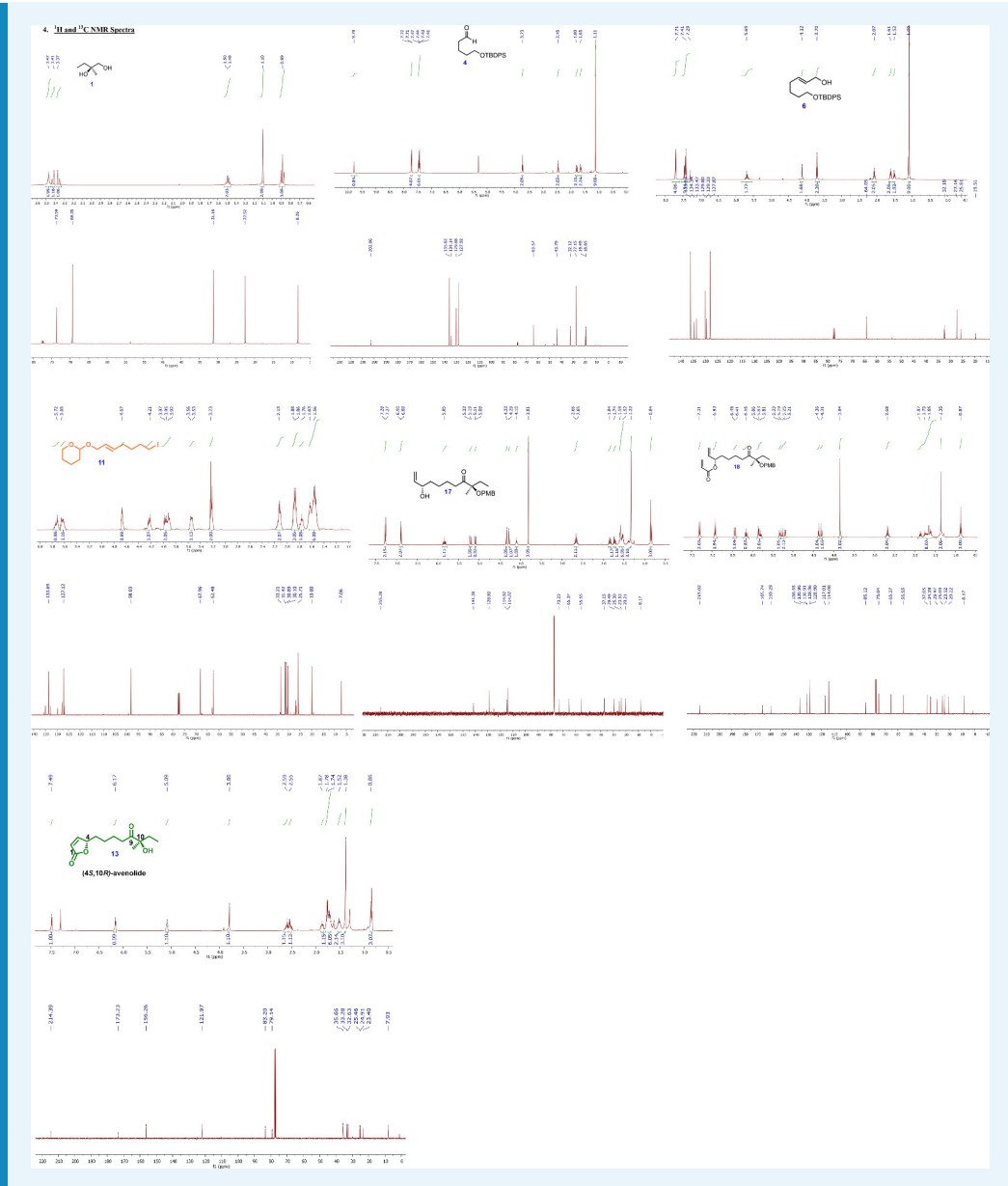

**Appendix 1—figure 3.** $^1$H and $^{13}$C NMR spectra of key intermediates and final product (4*S*,10*R*)-avenolide.

### 4. ¹H and ¹³C NMR Spectra

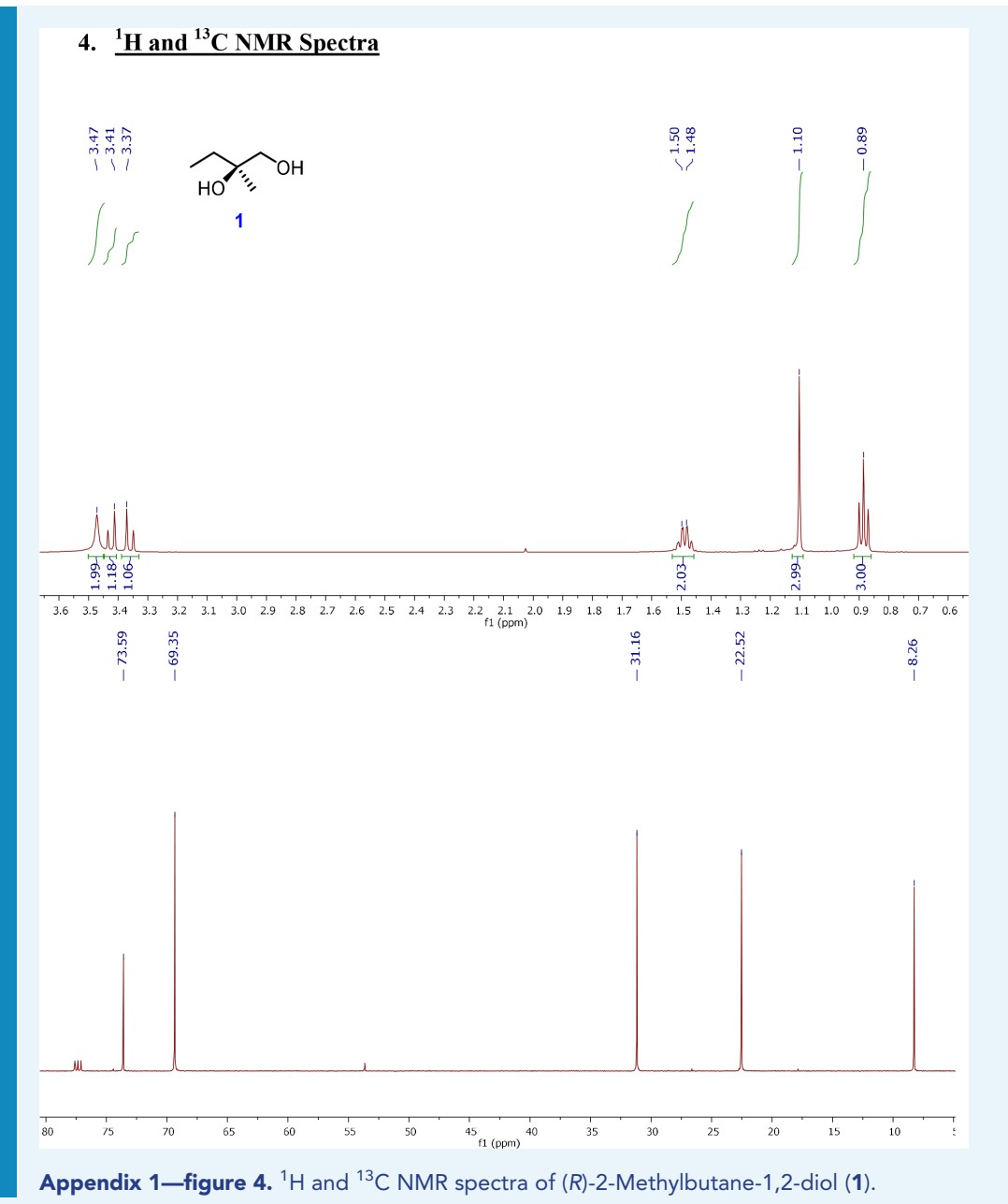

**Appendix 1—figure 4.** ¹H and ¹³C NMR spectra of (*R*)-2-Methylbutane-1,2-diol (**1**).

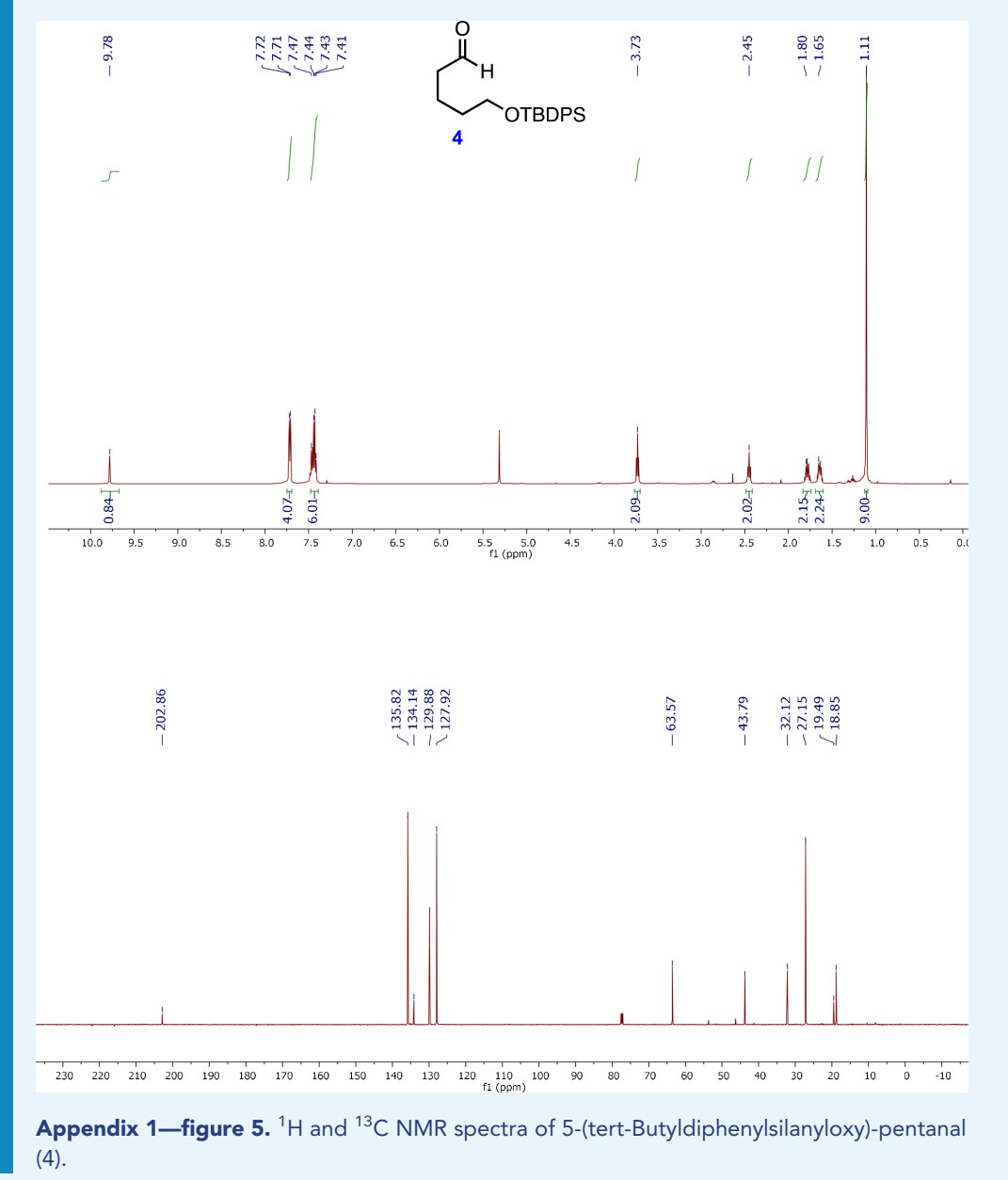

**Appendix 1—figure 5.** $^1$H and $^{13}$C NMR spectra of 5-(tert-Butyldiphenylsilanyloxy)-pentanal (4).

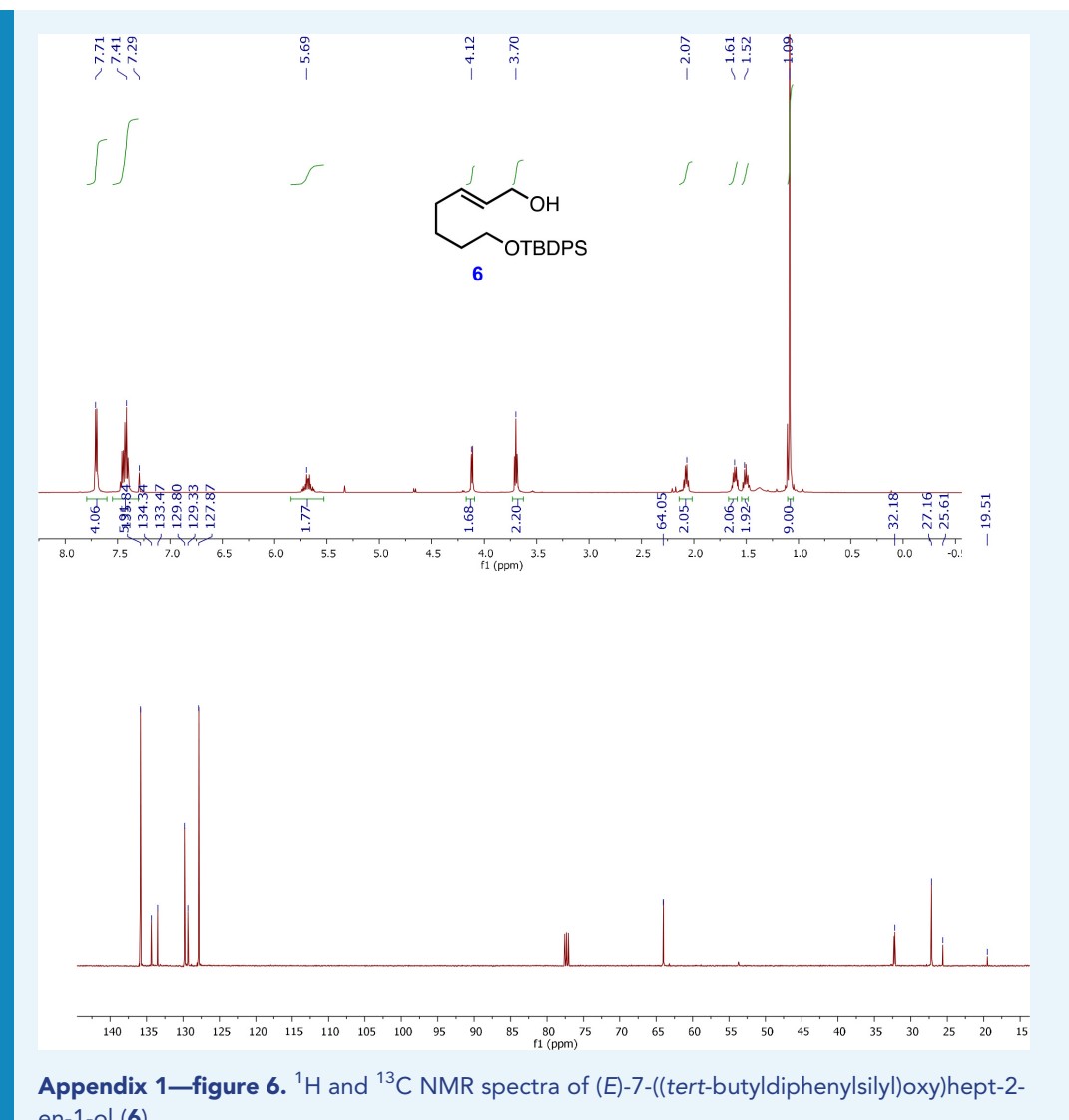

**Appendix 1—figure 6.** $^1$H and $^{13}$C NMR spectra of (E)-7-((*tert*-butyldiphenylsilyl)oxy)hept-2-en-1-ol (**6**).

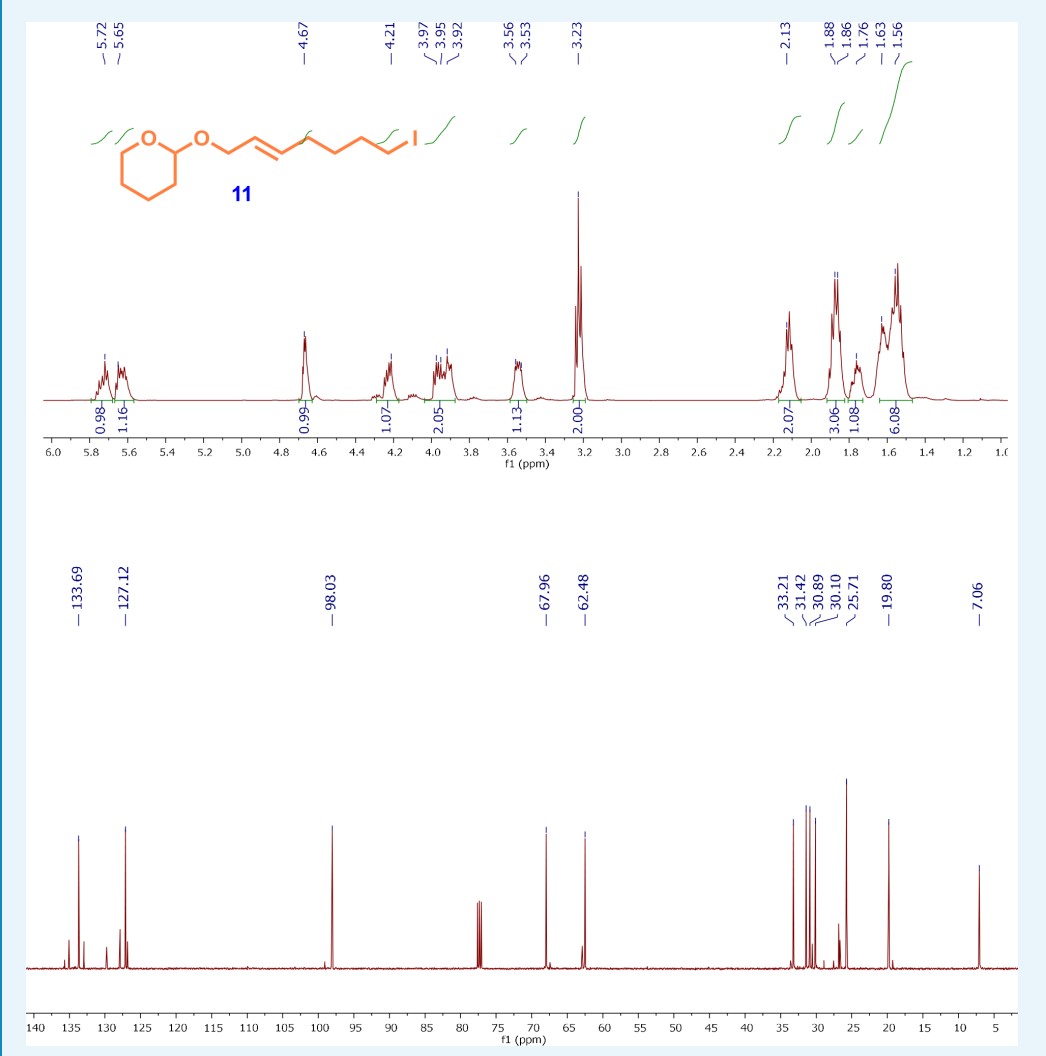

**Appendix 1—figure 7.** [1]H and[13]C NMR spectra of iodo-alkene intermediate; (E)-2-((7-iodo-hept-2-en-1-yl)oxy)tetrahydro-2H-pyran (11).

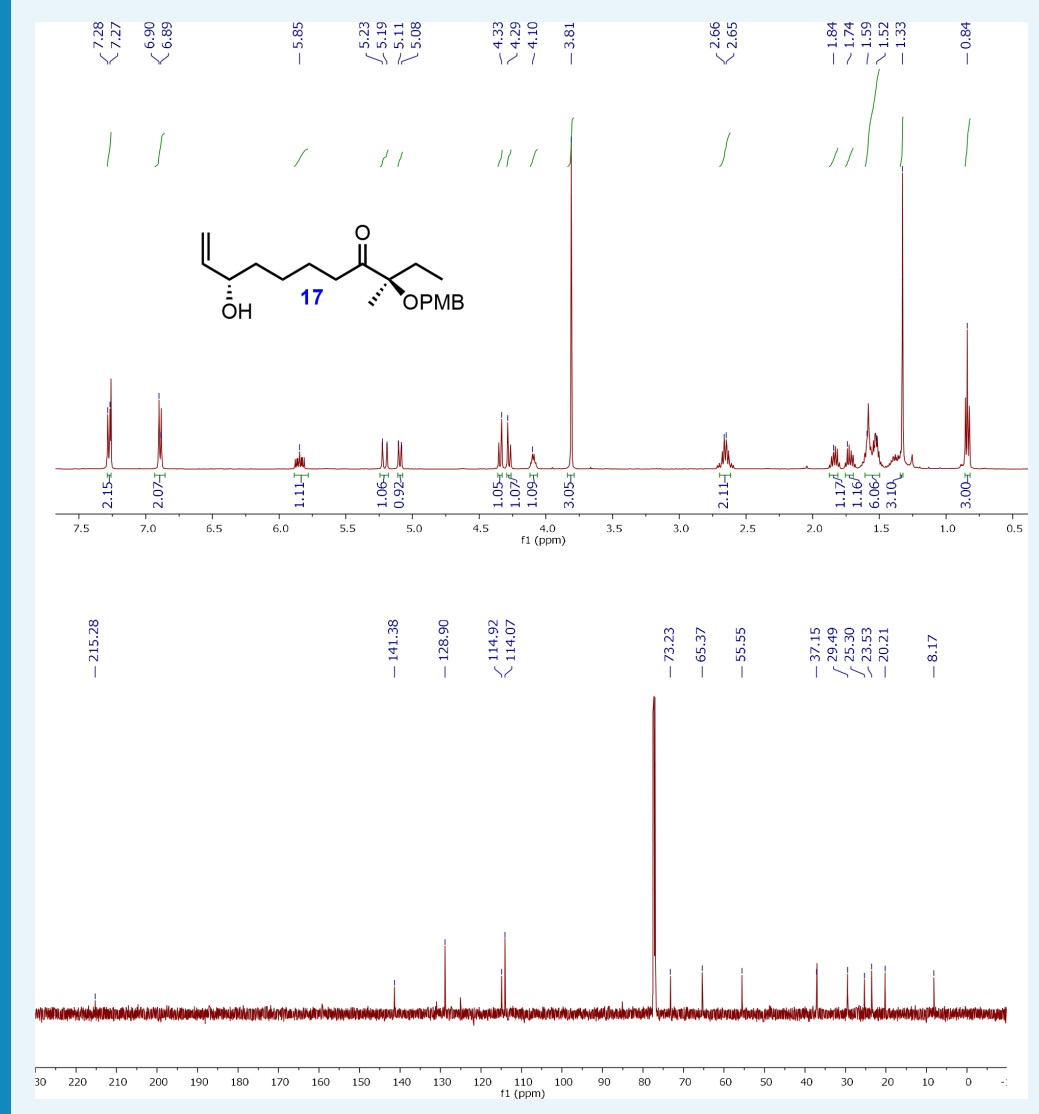

**Appendix 1—figure 8.** $^1$H and $^{13}$C NMR spectra of allyl alcohol intermediate; (*R*)-8-((2*S*,3*S*)3-(Hydroxymethyl)oxiran-2-yl)3-((4-methoxybenzyl)oxy)3-methyloctan-4-one (**17**).

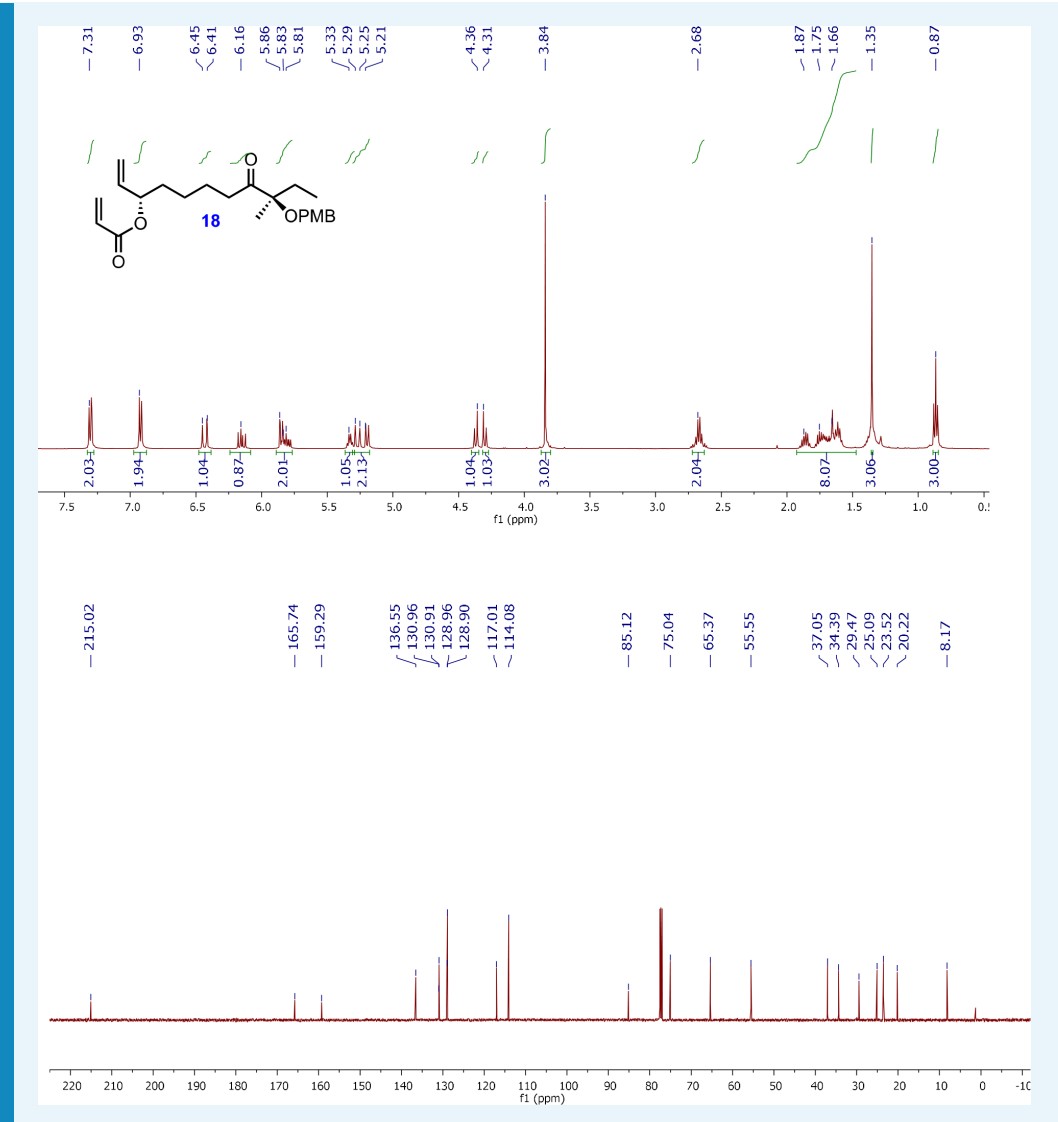

**Appendix 1—figure 9.** $^1$H and $^{13}$C NMR spectra of acrylyl alkene intermediate; (3*S*,9*R*)-9-((4-methoxybenzyl)oxy)-9-methyl-8-oxoundec-1-en-3-yl acrylate (**18**).

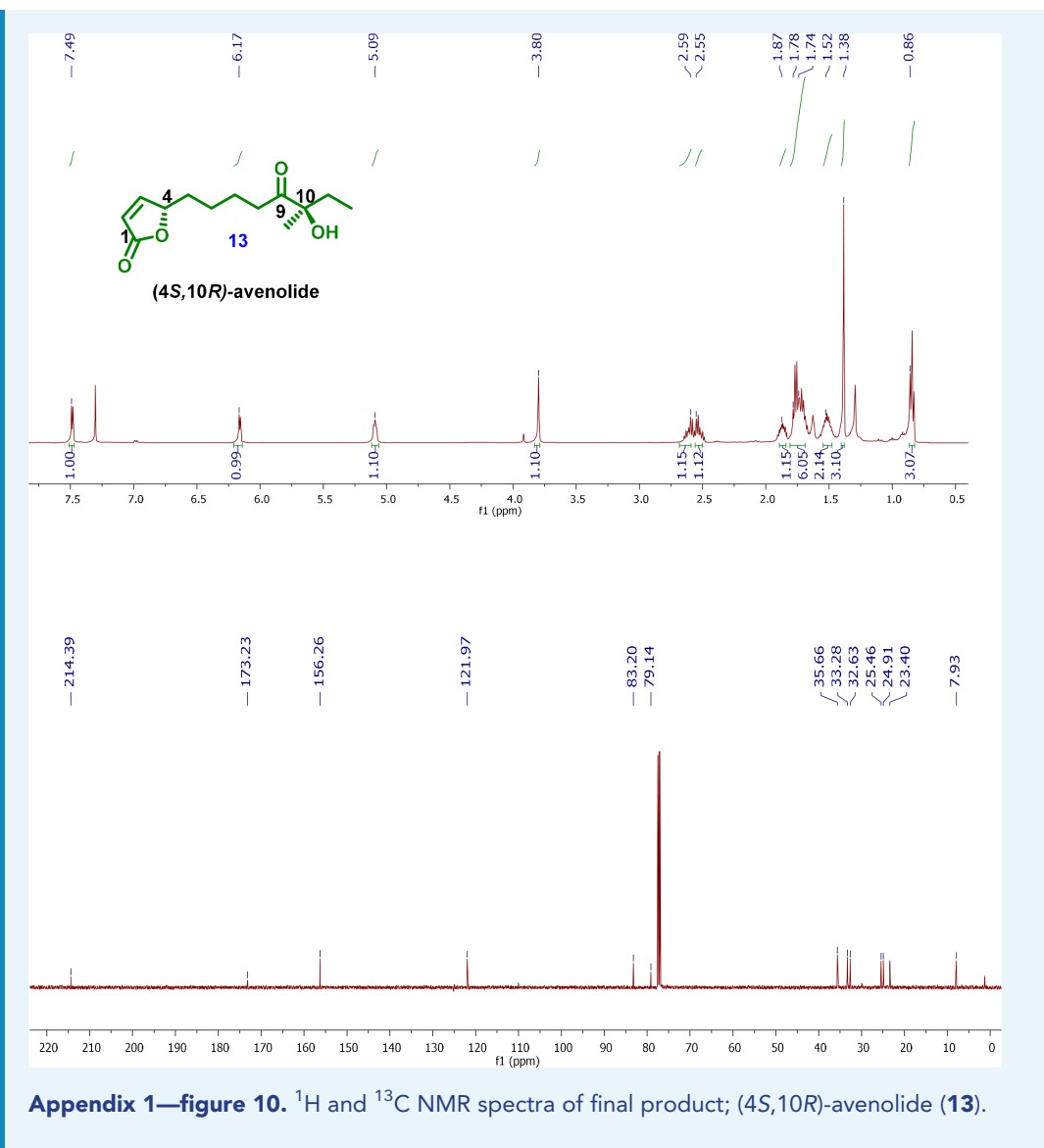

**Appendix 1—figure 10.** $^1$H and $^{13}$C NMR spectra of final product; (4S,10R)-avenolide (**13**).

