## [Decision Letter]

**Acceptance summary:**

This paper probes the molecular mechanism by which bacteria use small molecules to co-ordinately multicellular responses. The range of the study from total synthesis of complex small molecules to structural studies of the signaling system were all well done. There is potential for future research, possibly with identification of the upregulated small molecule(s)

**Decision letter after peer review:**

Thank you for submitting your article "Biochemical Basis for the Regulation of Biosynthesis of Antiparasitics by Bacterial Hormones" for consideration by *eLife*. Your article has been reviewed by three peer reviewers, including Jon Clardy as the Reviewing Editor and Reviewer #1, and the evaluation has been overseen by Michael Marletta as the Senior Editor. The following individual involved in review of your submission has agreed to reveal their identity: Brian Bachman (Reviewer #2).

The reviewers have discussed the reviews with one another and the Reviewing Editor has drafted this decision to help you prepare a revised submission.

Summary:

Humans use small diffusible molecules – hormones and neurotransmitters – to coordinate multicellular responses, and bacteria use a functionally equivalent system – bacterial hormones in this manuscript – to coordinate cellular behavior. This study investigated one foundational signaling system in molecular and mechanistic detail to produce an informative description of how the bacterial hormone avenolide binds to its cognate receptor AvaR1 to regulate the production of other diffusible small molecules, some of which could be previously undescribed and conceivably therapeutically useful. The report clearly describes the studies in convincing and clear detail.

Essential revisions:

Both peer reviewers and the Reviewing Editor thought that the article contained convincing data on an important topic and that a suitably revised manuscript would be a worthy contribution to *eLife*. They also felt that the manuscript needed additional work in explaining the motivation for some studies and ideally additional experimental data. Specifically:

• The manuscript described a detailed structural study on an important regulator of small molecule biosynthesis with potential significance to the discovery of new molecules that might be useful therapeutic leads. The manuscript concludes with a demonstration of this potential that is, to put it bluntly, underwhelming: color changes that might indicate the production of previously cryptic molecules. There is no additional evidence to support this interpretation. No differential metabolomics that would indicate production of a new molecule, no evidence that the production of any new molecule was induced through AvaR1 regulation, not even evidence that there was a qualitative, not quantitative, change in metabolism were presented.

• The manuscript would benefit from a clearer description of why avenolide was synthesized and why the strategy was significant. A 22-step synthesis is not something undertaken lightly. Was it synthesized because that was the most efficient way to get the desired ligand? Was it synthesized because eventually a synthesis combined with the structural data could lead to the design and synthesis of new activating ligands for different R-proteins?

Revisions expected in follow-up work:

In the event you are not able to provide more substantial proof of the ability/potential to discover new molecules to the current manuscript, we would suggest deleting the color change experiments. They by themselves actually undercut the significance of the present manuscript.

---

## [Author Response]

Essential revisions:Both peer reviewers and the Reviewing Editor thought that the article contained convincing data on an important topic and that a suitably revised manuscript would be a worthy contribution to eLife. They also felt that the manuscript needed additional work in explaining the motivation for some studies and ideally additional experimental data. Specifically:• The manuscript described a detailed structural study on an important regulator of small molecule biosynthesis with potential significance to the discovery of new molecules that might be useful therapeutic leads. The manuscript concludes with a demonstration of this potential that is, to put it bluntly, underwhelming: color changes that might indicate the production of previously cryptic molecules. There is no additional evidence to support this interpretation. No differential metabolomics that would indicate production of a new molecule, no evidence that the production of any new molecule was induced through AvaR1 regulation, not even evidence that there was a qualitative, not quantitative, change in metabolism were presented.

We thank the reviewers for their feedback and constructive criticisms of our manuscript. We had been able to obtain metabolomics data but could not conduct these in replicate prior to the shutdown. While these data do show evidence for the production of new (as yet unidentified) molecules, due to the lack of replicate data we have opted not to include them in the revised manuscript. We have deleted the color changing experiments as suggested by the reviewers.

• The manuscript would benefit from a clearer description of why avenolide was synthesized and why the strategy was significant. A 22-step synthesis is not something undertaken lightly. Was it synthesized because that was the most efficient way to get the desired ligand? Was it synthesized because eventually a synthesis combined with the structural data could lead to the design and synthesis of new activating ligands for different R-proteins?

We have added a clearer description of both the synthetic scheme and the reasoning for the synthetic approach. A synthetic approach was needed as it was the both the most efficient way to obtain the desired ligand, and because it may be easily adapted for the synthesis for other activating butenolides with slightly different chemical structures.

Revisions expected in follow-up work:In the event you are not able to provide more substantial proof of the ability/potential to discover new molecules to the current manuscript, we would suggest deleting the color change experiments. They by themselves actually undercut the significance of the present manuscript.

We agree and the color change experiments have been deleted.